# Palau's warmest reefs harbor thermally tolerant corals that thrive across different habitats

Hanny E. Rivera [1,2,3✉], Anne L. Cohen [2✉], Janelle R. Thompson[3,4,5], Iliana B. Baums [6], Michael D. Fox[2,7] & Kirstin S. Meyer-Kaiser[2]

Ocean warming is killing corals, but heat-tolerant populations exist; if protected, they could replenish affected reefs naturally or through restoration. Palau's Rock Islands experience consistently higher temperatures and extreme heatwaves, yet their diverse coral communities bleach less than those on Palau's cooler outer reefs. Here, we combined genetic analyses, bleaching histories and growth rates of *Porites* cf. *lobata* colonies to identify thermally tolerant genotypes, map their distribution, and investigate potential growth trade-offs. We identified four genetic lineages of *P.* cf. *lobata*. On Palau's outer reefs, a thermally sensitive lineage dominates. The Rock Islands harbor two lineages with enhanced thermal tolerance; one of which shows no consistent growth trade-off and also occurs on several outer reefs. This suggests that the Rock Islands provide naturally tolerant larvae to neighboring areas. Finding and protecting such sources of thermally-tolerant corals is key to reef survival under 21st century climate change.

[1] MIT-WHOI Joint Program in Oceanography/Applied Ocean Science & Engineering, Cambridge and Woods Hole, MA, USA. [2] Woods Hole Oceanographic Institution, Woods Hole, MA, USA. [3] Massachusetts Institute of Technology, Cambridge, MA, USA. [4] Asian School of the Environment, Nanyang Technological University, Singapore (NTU), Singapore. [5] Singapore Centre for Environmental Life Sciences Engineering (SCELSE), Singapore, Singapore. [6] Pennsylvania State University, State College, PA, USA. [7] Red Sea Research Center, King Abdullah University of Science and Technology (KAUST), Thuwal, Saudi Arabia. ✉email: hrivera28@gmail.com; acohen@whoi.edu

Ocean warming with increased marine heatwave intensity is considered the most significant threat to coral reefs globally[1]. Seawater temperatures >1 °C above historical summertime values can disrupt the coral-algae endosymbiotic relationship, leaving corals in a nutritionally compromised, pale state ("bleached") and vulnerable to death[1]. Since the early 1980's, millions of corals have died and thousands of acres of coral reef area have been lost as bleaching events have become more frequent, severe[2], and widespread[3].

As anthropogenic $CO_2$ emissions continue to rise, fueling further warming and more intense heatwaves, science and conservation efforts are increasingly focused on identifying thermally-tolerant coral communities that could survive ocean warming and potentially reseed impacted reefs either naturally or via restoration[4]. Environments with high variability are promising coral source sites because they may promote colony plasticity and harbor resilient communities[5–7]. In addition, habitats that currently experience conditions akin to those projected under climate change (e.g., higher temperatures, lower pH), are potential reservoirs of environmentally tolerant coral populations that may facilitate coral survival through the influx of climate-adapted offspring (i.e., evolutionary rescue[8]).

The Palauan archipelago harbors robust coral communities with a demonstrated tolerance to high temperatures[9–12]. South of Palau's mainland, hundreds of small islands—the Rock Islands—form a series of semi-enclosed bays. There, water temperatures are consistently higher, sometimes by as much as 2 °C, than on many of Palau's fringing, patch, and barrier reefs (hereafter called "outer reefs"; Fig. 1). Despite warmer temperatures, the Rock Islands have also maintained consistently high coral genera diversity and stable species compositions for 15 years (2002–2016), while genera diversity in many outer reefs suffered steep declines after the 2010 bleaching event[13]. In addition, there was higher coral cover (~60%) in the Rock Islands than on Palau's outer reefs (measured in 2010–2012)[14,15].

The Rock Island's warmer temperatures and high coral diversity may facilitate selection for communities with higher thermal tolerance. During heatwaves associated with the El Niño Southern Oscillation (ENSO), some Rock Island sites have experienced larger temperature anomalies than outer reefs[11,12]. For instance, during the 2010 ENSO, several of the Rock Island sites in this paper experienced anomalies of up to 2 °C, while outer reefs experienced either anomalies that were negative or

up to only 1 °C, relative to a 2001–2011 average[11]. During the 1997–1998 ENSO (the most devastating bleaching in Palau[9,10]) only ~25% of Rock Island corals bleached, compared to nearly 60% of corals in Palau's outer reefs; and Rock Island reefs also recovered faster[9,10,12]. This pattern was repeated during the shorter and less severe 2010 ENSO, when Rock Island temperatures were ~0.6 °C warmer than on outer reefs, but corals showed lower bleaching: ~15% vs. 25–30% on outer reefs[11,12]. Rock Island corals are also growing in low pH waters (as low as 7.8[15]) with moderate levels of turbidity and shading from surrounding vegetation[14,16]. In contrast, the outer reef sites experience near-open ocean conditions[15]. Reduced ENSO-associated bleaching at the Rock Islands is notable since compounding heat and pH stress often leads to more severe bleaching across many coral species[17,18].

*Porites* cf. *lobata*, is a genetically diverse, ubiquitous species on Indo-Pacific coral reefs[19–21], and also common across Palau's reef habitats, from the semi-isolated Rock Island bays to the most exposed reefs ("cf." denotes species identification uncertainty, in this case from morphological plasticity and potentially cryptic species[19,20]). *Porites* corals are more resilient to bleaching than other species, in part because they can utilize heterotrophic food sources and their polyps can retract deep within the skeleton[22]. Colonies can live for hundreds of years and grow to be meters in diameter and can form microatolls with valuable microhabitats for a variety of fish and invertebrate species[23–25]. Given its tolerance of marginal environments, its role as habitat for other reef organisms and its resiliency, *P.* cf. *lobata* is a key species for understanding the future of coral reefs.

Furthermore, skeletons of mounding, long-lived corals like *P.* cf. *lobata* contain quantitative information about their annual extension (upward growth), skeletal density, and calcification rates that can be measured from computed tomography (CT) scans of skeletal cores. During bleaching, skeletal extension diminishes and corals form anomalous, high-density "stress bands" that can be detected in the same CT images to track the bleaching history of individual corals[12,26–34]. The formation of these stress bands is likely linked to bleaching-induced starvation and reduced linear extension[31,32]. Analyzing stress band histories during periods of known thermal stress can therefore provide insights into the thermal sensitivity of individual colonies and among reefs under natural conditions[30,31]. Prior work in Palau has shown that Rock Island *P. lobata* formed fewer stress bands

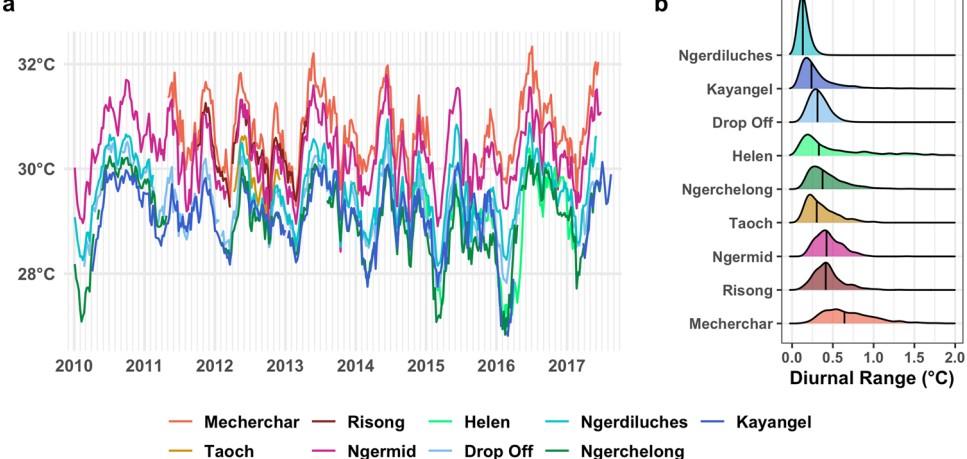

**Fig. 1 Water temperatures across Palau's reefs.** In situ logger temperatures with Rock Island sites shown in warm (red/orange) tones and outer reef sites in cool (blue/green) tones; see Fig 2 for map. **a** Weekly averaged time series by site. **b** Distributions of diurnal ranges (max-min daily temperature) across each site (temperature time series were season- and year-adjusted by band-pass filtering to remove long (>36-h) and short (<5 h) variability prior to calculating diurnal range). Vertical lines represent distribution median values.

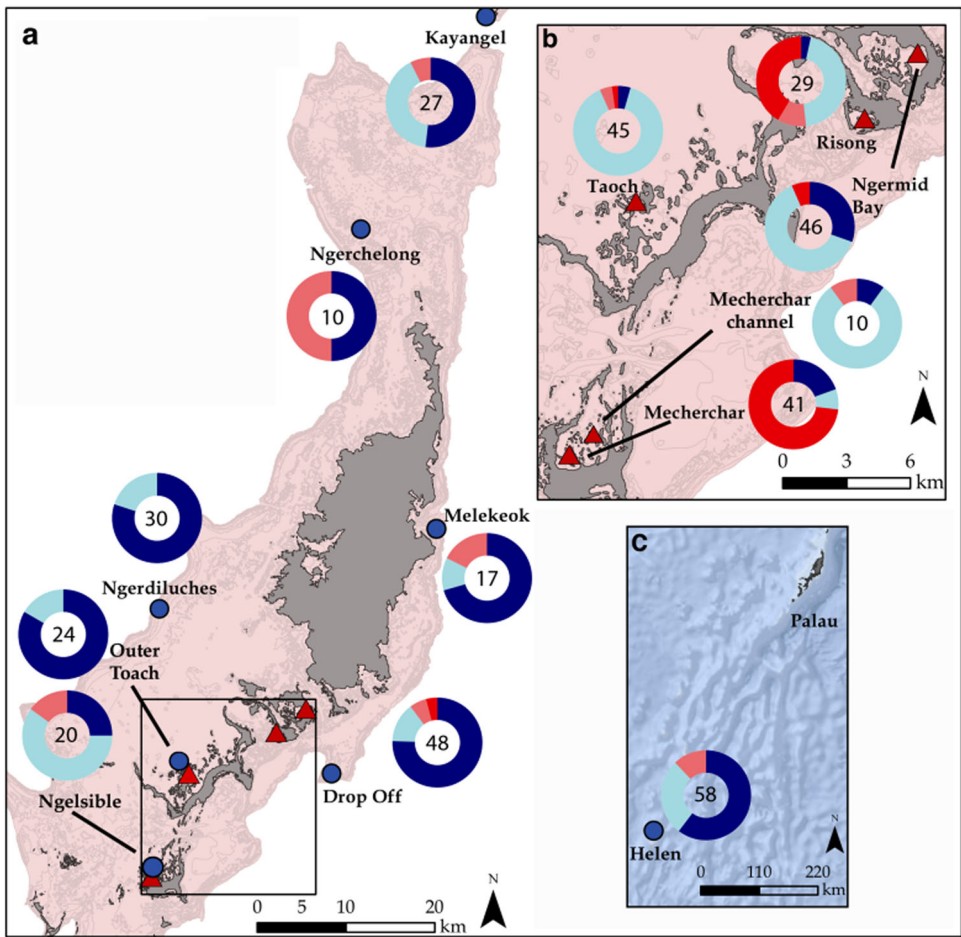

**Fig. 2 Distribution of *Porites* cf. *lobata* lineages across sampling sites.** Land is denoted in gray and the reef platform in light pink. Outer reef sites are denoted by blue circles, while Rock Island sites are denoted by red triangles. Donut charts show the distribution of lineages among samples from each site where coral lineages are color coded Dark Blue (DB), Light Blue (LB), Pink (Pl), and Red (RD) for visualization. Charts include both RAD-seq and microsatellite data. (For samples with both types of data, RAD-seq derived lineages were used as these predominantly agreed with microsatellite results, see Methods section and Fig. S2). Total *N* for each site is shown in the center of each donut chart. **a** Palauan mainland. **b** Inset showing Rock Island sites. **c** Palau and Helen atoll (Palau's southernmost territory). Reef line shape files were accessed from the NOAA National Centers for Coastal Ocean Science (https://maps.coastalscience.noaa.gov/biomapper/biomapper.html?id=Palau) and incorporated into landmass basemaps available in ArcMap™.

during the 1998 and 2010 ENSO events than *P. lobata* living on the outer reefs[12] (i.e., they are more thermally tolerant). While harboring more thermally tolerant genera of Symbiodiniceae, such as *Durusdinium*, can make corals less susceptible to bleaching[35], massive and branching *Porites* species throughout the Indo-Pacific nearly exclusively harbor *Cladocopium* (formerly C15)[36–38], with symbionts transmitted vertically from mother to offspring[39]. This makes *P.* cf. *lobata* an ideal study system, as its symbiont fidelity allows one to investigate coral host response and adaptation, while its mounding morphology enables quantification of historical growth and bleaching responses under naturally occurring heatwaves.

The existence of healthy coral communities with higher tolerance for and resilience to thermal stress within Palau's Rock Island habitats raises key questions about the drivers of thermal tolerance. Here, we ask if Rock Island *P.* cf. *lobata* are genetically distinct from their outer reef counterparts (e.g., represent different lineages or potentially cryptic species), and whether any such distinct genetic groups are restricted to Rock Island habitats. We further couple genetic data with colony-specific temperature tolerance (presence/absence of stress bands during prior heatwaves) and skeletal traits (density, linear extension, and calcification rates) to examine if: (1) thermal tolerance differs among

genetic lineages; and (2) whether thermal tolerance leads to trade-offs in growth. Using 12 microsatellite markers and 12,761 single nucleotide polymorphism (SNP) loci generated from Restriction-site Associated DNA (RAD) sequencing, we explore the genetic structure of *P.* cf. *lobata* colonies across five Rock Island and eight outer reefs (Fig. 2). To our knowledge, our study is the first to combine individual colony genetics with growth parameters and bleaching histories of responses to natural ENSO heatwaves through coupled genetic sampling and coral coring. We find four distinct genetic lineages of *P.* cf. *lobata* across Palau with varying thermal tolerance.

## Results

**Rock Island sites have consistently warmer temperatures and a higher diurnal range.** Multiple years (2010–2018) of weekly averaged in situ temperature data from four Rock Island sites show these locations are consistently warmer than all outer reef sites (Fig. 1a). Across Palau's outer reefs, average water temperatures were 29.11 °C ±0.7 (standard deviation), with a mean diurnal range of 0.32 °C. Mean temperatures on Rock Island reefs were ~1.5 °C warmer at 30.29 °C ±0.62, and with an 85% higher mean diurnal range of 0.59 °C. Some days at Ngerchelong and

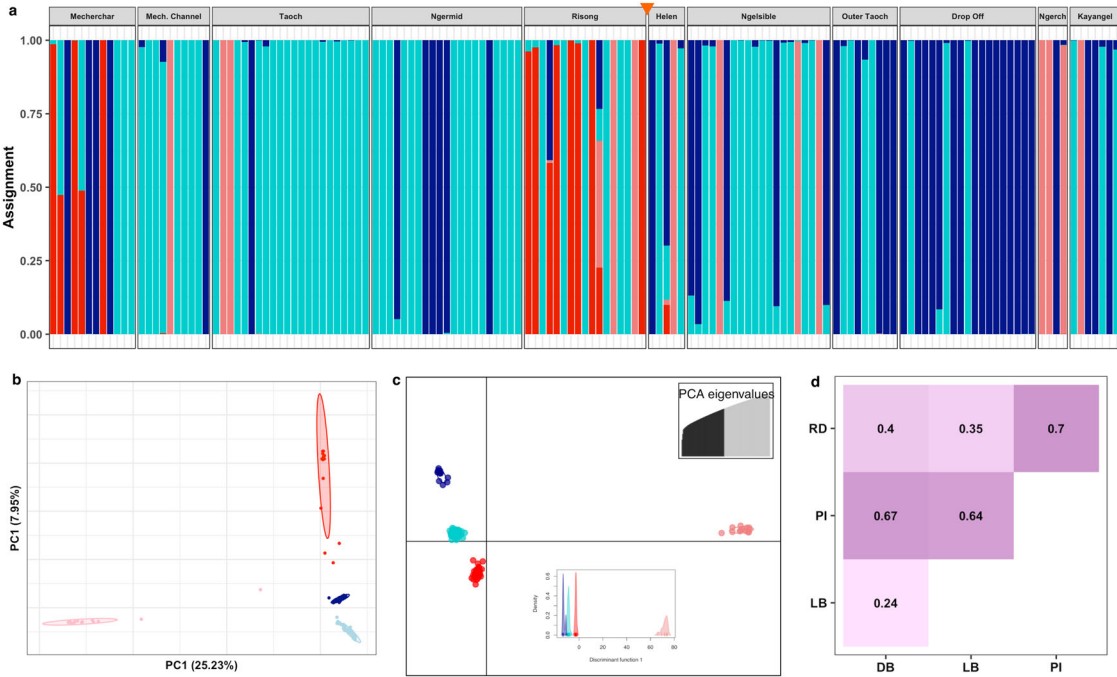

**Fig. 3 RAD-seq based population structure of *Porites* cf. *lobata*. a** STRUCTURE for $K = 4$. Rock Island sites are shown first (left of orange triangle), followed by outer reef sites (right of triangle). RI sites are dominated by light blue (LB) and red lineages (RD) while the dark blue (DB) and pink lineages (PI) are more common on the outer reefs. **b** Principal component analyses of RAD-seq data. Points represent individual samples and are colored by their dominant lineage (>50% assigned lineage based on STRUCTURE results). **c** Discriminant analysis of principal components (DAPC) recapitulates the four lineage clusters. Points are individual samples, colored by their dominant lineage assignment. Top inset shows number of retained principal components for analysis (70) based on cross-validation optimization. Lower inset shows density across the first discriminant axis. **d** Pairwise Nei's $F_{ST}$ values between lineages. Background color intensity increases with higher $F_{ST}$.

Helen reefs (both outer reefs) had wide temperature ranges, but these were rare events compared to the more frequent high-range days at most other Rock Island reefs (Fig. 1b). Taking the 90th percentile warmest temperatures each year, the top 10% of peak temperatures in the Rock Islands were above 31.08 °C on average, compared with only 29.78 °C on the outer reefs. These in situ temperatures help support prior temperature studies in Palau that found long-term differences in temperature patterns across the Rock Islands and outer reefs[9,11].

***Porites* cf. *lobata* in Palau form four distinct genetic lineages across habitats with varying temperatures, and lineages host the same symbiont types**. Microsatellite genotyping and RAD-sequencing analyses revealed that *P.* cf. *lobata* populations across the Palauan archipelago form four distinct lineages with high genetic divergence (Figs. 2 and 3 and S1). STRUCTURE assignments of samples to genetic lineages were consistent between microsatellites and RAD-seq datasets, with 96% of samples assigning to the same lineage (Fig. S2). PCA and DAPC analyses supported the presence of four strongly differentiated lineages in RAD-seq (Fig. 3b, c) and microsatellite data (Fig. S1B, C). Please note, the color designation of lineages in figures is simply for visualization purposes and does not reflect differences in colony color.

As expected, $F_{ST}$ values between lineages were higher when calculated based on RAD-seq rather than multi-allelic microsatellite data[40], with ranges between 0.24 and 0.67 for RAD-seq data and 0.07 to 0.16 for microsatellite data (Figs. 3d and S1D). The ordering of pairwise differences remained the same between the two datasets. The dark blue (DB) and light blue (LB) lineages were the least differentiated (RAD $F_{ST} = 0.24$; microsatellite $F_{ST} = 0.07$), and had more admixed individuals (Figs. 3a and

S1A). See Supplementary Data 1 for pairwise $F_{ST}$ values by site for each lineage, though note that for many sitewise comparisons, samples sizes were less than five (these are denoted in the dataset). The high (>0.3) $F_{ST}$ values and the high lineage assignment probabilities suggests that some of these lineages could represent cryptic species of *Porites lobata* or other massive *Porites* species across the Palauan archipelago. The presence of several admixed individuals (Figs. 3a and S1A) is consistent with ongoing hybridization and/or past introgression, though admixture was uneven among lineages. We have observed synchronous spawning and successful cross-fertilization for at least two lineages from Rock Island and outer reef sites, making ongoing hybridization likely (Meyer-Kaiser et al., personal observations, 2022). The lineage with the highest $F_{ST}$, pink (PI), also had the smallest sample sizes ($N = 11$ for RAD-seq; $N = 20$ for microsatellites), which may be inflating $F_{ST}$ values. Most of the PI samples are from Kayangel, Helen, Melekeok, and Ngerchelong, all fairly distant sites (Figs. 2 and 3). As such, the high between-lineage $F_{ST}$ could be inflated by isolation-by-distance effects. Here, we refer to these genetic groups as differentiated lineages, absent solid evidence of reproductive isolation or incompatibility that would support reclassification based on the biological species concept.

The four lineages were differentially distributed across Palau's reef habitats, with some lineages predominantly found in the warmer Rock Islands, while others were dominant on cooler outer reefs (Fig. 2). The DB lineage was predominant among outer reefs, while the LB lineage was the most widespread, though it occurred in higher proportion within warmer Rock Island sites (Fig. 2). The PI lineage was more common on outer reefs but represented a small fraction of the community at most sites, except in Ngerchelong (Fig. 2). The red (RD) lineage was entirely confined to the Rock Islands, except one individual from Drop

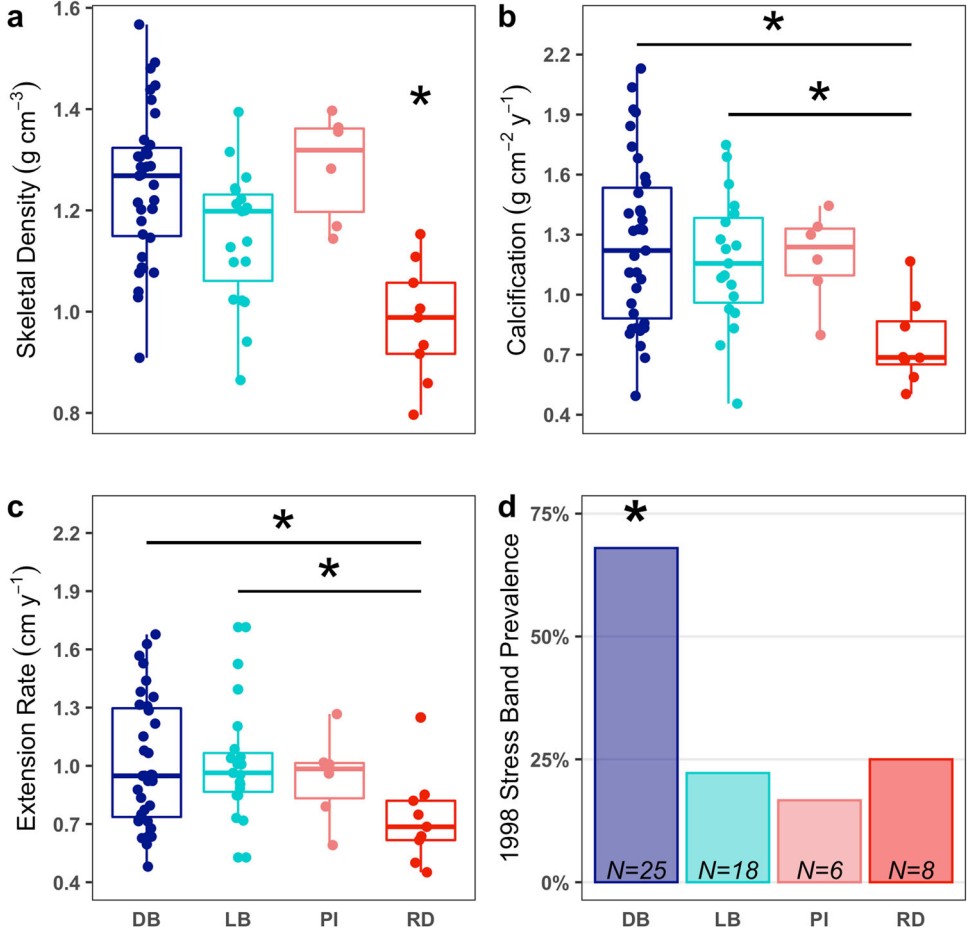

**Fig. 4 Growth and thermal tolerance of corals from different lineages based on analysis of individual coral cores.** In **a–c**, line segments and asterisks denote statistically significant differences ($p < 0.05$) between groups based on post hoc Tukey tests. An asterisk over a single group denotes that group is significantly different from all others. **a** Boxplots of mean skeletal density of samples from each lineage. The Red (RD) lineage had lower skeletal density than all other lineages (ANOVA, $F = 10.73$, df = 3, $p < 0.001$). **b** Boxplots of mean calcification rate of samples from each lineage. RD showed lower calcification rates than the DB and LB lineages (ANOVA, $F = 5.36$, df = 3, $p < 0.05$). **c** Boxplots of mean extension rate of samples from each lineage. RD showed lower extension rates than the DB and LB lineages (ANOVA, $F = 2.97$, df = 3, $p < 0.05$). **d** Stress band prevalence by lineage in 1998. Proportion of stress differed significantly among lineages with DB lineage showing higher prevalence than all other lineages ($X^2 = 12.349$, df = 3, $p$ value < 0.05). Note that $N$ values in **d** are lower than in **a–c** as not all cores extended back to 1998 (see Methods for further details).

Off (Figs. 2 and 3a). Within the Rock Islands, RD corals were common in Mecherchar and Risong but rare elsewhere (Fig. 2).

We found no differences in the symbiont composition of colonies using denaturing gel electrophoresis (DGGE) and ITS2 sequencing (Supplementary Table 1). In fact, all colonies harbored *Cladocopium* (C15) symbionts, which are strongly associated with massive and branching *Porites* corals across the Indo-Pacific, including Palau[36–38]. Looking at 218 RAD-seq derived SNPs across 137 individuals, we find that the symbionts cluster into distinct groups that recapitulate their coral host lineages, as would be expected for vertically transmitted symbionts (Fig. S3). Genetic differentiation between symbiont clusters was much lower than in the coral host, however, with $F_{ST}$ values of <0.1 for most comparisons, with the exception of the PI lineage (Fig. S3C).

**Lineages had different growth rates and thermal tolerances.** Skeletal density, calcification, and linear extension rates differed among lineages (Fig. 4a–c). The Rock Island-associated RD lineage had lower skeletal density than all other lineages (ANOVA, $F = 10.73$, df = 3, $p < 0.001$; with post hoc Tukey test, $p < 0.05$; Fig. 4a), as well as lower calcification and extensions

rates than the DB and LB lineages (ANOVA, $F = 5.36$, $F = 2.97$, respectively, df = 3, $p < 0.05$; with post hoc Tukey tests, $p < 0.05$; Fig. 4b, c). The outer reef-associated PI lineage did not have significantly lower growth rates for any metric compared with the DB and LB lineages, but it was also not significantly different from the RD linage in calcification or extension rates, showing an intermediate phenotype between the DB and LB lineages and RD (Fig. 4a–c).

The presence or absence of high-density stress bands in mounding *Porites* cores provides valuable insight to differences in thermal tolerance among individuals and across habitats (e.g., Fig. S4). The widespread outer reef DB lineage showed the highest prevalence of stress bands (68%) during the 1998 ENSO, suggesting this lineage has low thermal tolerance (Fig. 4d). In contrast, the Rock Island-associated LB and RD lineages showed significantly lower stress band prevalence at 22% and 25%, respectively ($X^2 = 12.35$, df = 3, $p$ value = 0.006; Fig. 4d). The PI lineage also showed low stress band prevalence, though we had the fewest number of samples from this lineage ($N = 6$). During the less severe 2010 ENSO, lineages showed overall lower prevalence of stress bands across all lineages (Fig. S5). Notably, LB and DB had similar growth rates and skeletal densities despite their differences in thermal tolerance in 1998 (Fig. 4). The RD

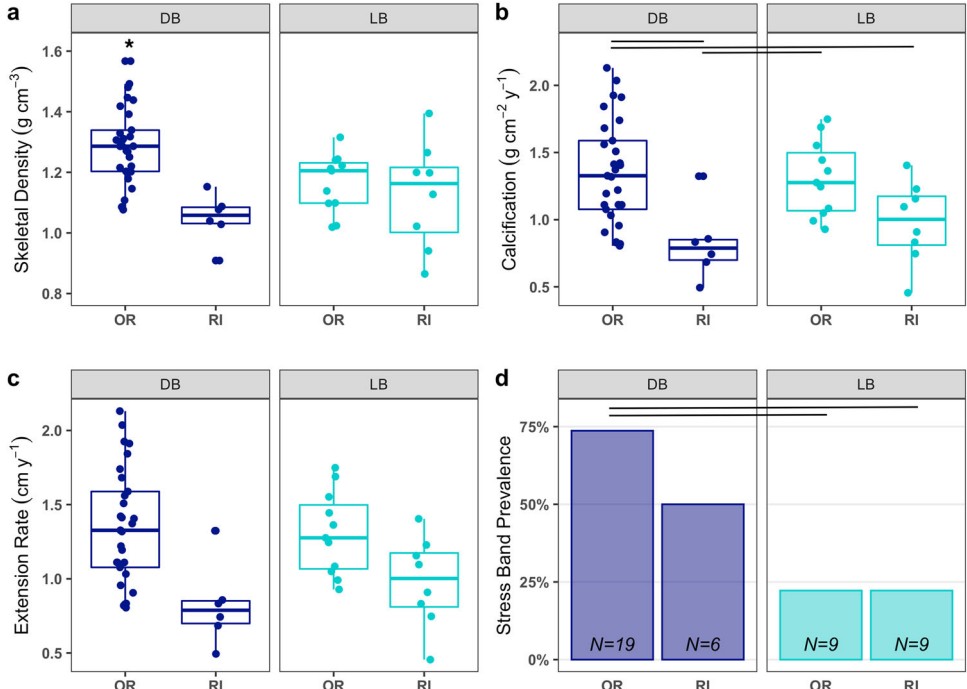

**Fig. 5 Growth and thermal tolerance of corals from LB and DB lineages based on analysis of individual coral cores distinguished by habitat region.** Outer reef (OR) and Rock Island (RI). In **a–c**, line segments denote statistically significant differences ($p < 0.05$) between groups based on post hoc Tukey tests. An asterisk over a single group denotes that group is different ($p < 0.05$) from all others. **a** Boxplots of mean skeletal density of samples from each lineage and region. The DB lineage in the outer reefs shows higher density than all other groups (ANOVA, $F = 9.212$, df = 3, $p < 0.001$). DB density in RI habitats is significantly lower than in the ORs (1.05 vs. 1.298 g cm$^{-3}$; Post hoc Tukey $p < 0.001$). Density in LB corals does not differ among habitats. However, they are lower than the DB outer reef coral density in both habitats ($p < 0.05$). **b** Boxplots of mean calcification rate of each lineage. There are significant differences among lineages and habitats (ANOVA, $F = 5.498$, df = 3, $p < 0.01$). The DB lineage shows significantly lower calcification in the Rock Island habitats compared to the ORs (1.35 vs. 0.82 g cm$^{-2}$ year$^{-1}$; Post hoc Tukey $p < 0.01$). DB corals in the ORs also have significantly higher calcification than LB corals in the RI ($p = 0.04$). LB corals in the ORs also have higher calcification than DB corals in the RI ($p = 0.04$). **c** Boxplots of mean extension rate of samples from each lineage. ANOVA showed significant differences among lineage and habitats ($F = 3.229$, df = 3, $p = 0.03$), but there were no significant pairwise differences in the post hoc test (difference between outer reef LB and Rock Island DB was marginal, $p = 0.053$). **d** Stress band prevalence by lineage in 1998. The DB lineage shows a trend toward lower stress band prevalence in the RI habitats ($X^2 = 0.34$, df = 1, $p = 0.56$). The LB lineage does not show any significant difference in stress band prevalence ($X^2 = 0$, df = 1, $p = 1$).

lineage demonstrated high thermal tolerance and low skeletal growth and density, but these corals are confined to two of the Rock Island sites with the lowest pH[14,15,41], which can slow growth and reduce density[42,43] (Figs. 2 and 4).

As DB and LB lineages were found across both Rock Island and outer reef sites in sufficient numbers for analysis, we compared growth and thermal tolerance metrics across habitats (Fig. 5). There were significant differences between site-lineage groups for all growth parameters (one-way ANOVAs for skeletal density (df = 3, $F = 9.21$, $p < 0.001$), calcification rate (df = 3, $F = 5.49$, $p < 0.01$), extension rate (df = 3, $F = 3.22$, $p < 0.05$)). The outer reef-associated DB corals had significantly lower skeletal density and calcification rate when living in the Rock Islands compared to outer reefs (Fig. 5a–c; pairwise Tukey tests for density ($p < 0.001$) and calcification rate ($p = 0.007$)), but the difference in extension rate was not significant ($p = 0.11$). Meanwhile, the Rock Island-associated LB corals did not show any significant differences between Rock Island and outer reef sites for any growth measures (Fig. 5a–c; pairwise Tukey tests for density ($p = 0.90$), calcification rate ($p = 0.19$), and extension rate ($p = 0.17$)). LB corals maintained low stress band prevalence in both Rock Island and outer reefs, maintaining thermal tolerance regardless of habitat type (Fig. 5d; $X^2 = 0$, df = 1, $p = 1$). Thus, our data showed clear differences in growth and thermal tolerance for DB corals in different environments but not for LB.

## Discussion

Here we are the first, to our knowledge, to combine historical growth and bleaching responses to ENSO heatwaves in the field (from coral cores) with genetic data. We find that Palau's *Porites* cf. *lobata* populations form distinct genetic lineages with differing thermal tolerances (Figs. 2–4). The unique environmental conditions of Palau's Rock Islands, where higher temperatures can serve as selective pressure and larger temperature ranges can facilitate plasticity, have likely promoted the development of thermally tolerant lineages (Fig. 1). Lineages most common in the Rock Islands can tolerate higher temperatures before bleaching and bleach less frequently (Fig. 4). When these lineages are found outside of Rock Island habitats, they maintain higher tolerance, suggesting this tolerance is at least partially driven by genetic factors in the host and possibly the symbiont (Figs. 5d and S3). Furthermore, the Rock Island-associated LB lineage maintains higher thermal tolerance across habitats, without showing any corresponding trade-offs in growth metrics (Fig. 5a–c). Our results expand upon prior work examining the influence of warmer and/or more variable habitats in promoting thermal tolerance (e.g., the back reef pools of Samoa[6]; the Florida reef tract[7,44,45], and Australia[46,47]) to show that these patterns are repeatable across multiple reefs. Our results also reinforce the notion that diverse habitats can yield climate change-resistant corals[48].

Several mechanisms could contribute to the enhanced thermal tolerance of Rock Island corals. First, the higher and more variable temperatures can serve as a filter, selecting for thermally tolerant corals while selecting against less thermally tolerant larvae arriving from outer reefs. Differential mortality among individuals in the early life-history stages could drive the population differentiation seen here (Figs. 2 and 3). The post-settlement period is a strong bottleneck with high mortality in many sessile invertebrates[49]. In fact, our samples included some small corals that are likely only a few years old, suggesting that selection acts very early in life-history, either on larvae or newly-settled recruits. In addition, limited connectivity between the Rock Island and outer reefs as suggested by hydrodynamic model output[50–52] could prevent larvae from dispersing away from their natal site and restrict gene flow among populations. The estimated retention time for one Rock Island site, Ngermid, is much longer than the larval duration for most corals, suggesting limited dispersal[51].

Local adaptation likely drives selection and population differentiation for *P.* cf. *lobata*. We saw strong genetic differentiation between lineages and found that reefs within the Rock Islands have different lineage compositions than outer reefs (Figs. 2 and 3 and S1). The outer reef-associated DB lineage showed significantly higher bleaching (lower thermal tolerance) than other lineages during the 1998 ENSO event, while the LB and RD lineages, most common in the Rock Islands, had significantly lower bleaching (Fig. 4d). These patterns are consistent with the notion that Rock Island environments select for lineages that are adapted to tolerate warmer temperatures. This is further supported by the LB lineage maintaining low bleaching prevalence in 1998 when living on the outer reefs, which points to a genetic basis for its thermal tolerance (Fig. 5d).

Second, daily temperature ranges between 0.5 to 5 °C have been found to be a better predictor of bleaching patterns than other environmental parameters[5], and in both natural and experimental studies, corals from variable environments often show increased thermal tolerance[6,47,53]. For instance, in American Samoa, *Acropora hyacinthus* from a highly variable back reef pool (~3–5 °C of diurnal range) showed higher thermal tolerance than conspecifics from an adjacent, less variable pool[6]. Rock Island reefs have a larger diurnal range (0.59 °C) than the outer reefs (0.32 °C; Fig. 1b), which could also facilitate thermal tolerance through plasticity mechanisms. In addition to Rock Island lineages being more thermally tolerant, outer reef-associated DB corals showed a trend toward fewer stress bands during the 1998 mass bleaching event when they had grown up and lived in the more variable Rock Islands (Fig. 5d). This suggests the variable Rock Island environments could also be promoting thermal tolerance via phenotypic plasticity. Alternatively, Rock Island conditions may have selected for the most thermally tolerant of the incoming DB larvae, or other Rock Islands conditions could mitigate bleaching in other ways. Additional studies that conduct reciprocal transplants of larvae and new recruits and robustly measure environmental variables would be needed to test hypotheses regarding the relative contributions of plasticity and adaptation in promoting thermal tolerance within the Rock Island habitats.

Lastly, symbiont co-evolution with particular lineages could also facilitate thermal tolerance. While all the hosted symbionts are the same species (*Cladocopium sp.* formerly clade C15), our RAD-seq symbiont derived SNPs do show strong symbiont fidelity and substructure corresponding to the coral host lineages (Fig. S3). The levels of divergence ($F_{ST}$), however, are much lower than between the coral host lineages (min $F_{ST}$ 0.04 in symbionts vs. 0.24 in corals; Figs. 3d and S3C), suggesting that coral genetic differences likely play a stronger role in the phenotypic differences observed between lineages. Nevertheless, the potential influence of symbiont lineages or a tight co-evolution between the

symbiont and host cannot be ruled out. Culture experiments that test the thermal tolerance of symbionts isolated from different host lineages would help elucidate the contribution of the symbionts to thermal tolerance patterns across Palau.

Future studies can also help determine the genetic underpinnings of the thermal tolerance differences seen across different lineages or elucidate other environmental contributors to the higher thermal tolerance of RI corals. For instance, Rock Island habitats differ from outer reefs in other parameters such as flow and light levels[11,36,54]. Shading in particular has been posited as a potential explanation for the thermal tolerance of Rock Island coral communities, as it may mitigate the irradiance stress that is often coupled with high temperature[11,55]. A lineage selected for lower light tolerance could appear thermally tolerant during heatwaves, as it may have experienced less stress residing in the shade. Such niche partitioning between lineages could both help drive sympatric divergence as well as help drive thermal tolerance (or the appearance thereof) if a particular lineage thrives best in microhabitats that may mitigate thermal stress. The LB lineage is most promising as a candidate to investigate a genetic basis for thermal tolerance in its own right because it maintains low bleaching levels across all habitats including unshaded outer reefs (Fig. 5). Adaptation to lower light levels could be a possible mechanism for the apparent thermal tolerance seen in the RD lineage, which is most common on one of the shadier Rock Island sites (Mecherchar; Fig. 2). Corals in Mecherchar grow mostly under the canopy of the trees along a narrow ledge (H. Rivera, personal observation). Of particular interest is the change in community composition seen between Mecherchar, which is RD dominated, and the nearby Mecherchar Channel site, which is LB dominated and where corals grow in the center of the pass, largely away from the shade of the nearby vegetation (Fig. 2). Lastly, a prior study using point measurements saw differences in nutrient levels between some RI sites and some outer reefs, with some RI sites showing higher nutrient levels[15]. The Rock Island environments, being closer to shore, may also differ in their supply of organic matter and facilitate increased heterotrophy. Prior studies have suggested that corals from nearshore reefs have more energy reserves[56], and in general increased heterotrophy has been shown to increase growth[57] or reduce bleaching response[58]. Understanding mechanisms of coral thermal tolerance would benefit from the kind of detailed, long-term environmental data beyond temperature that is more commonly available to terrestrial researchers but often lacking for marine environments.

Higher resolution of genetic and environmental data in coral systems will also help elucidate drivers of strong genetic divergence between sympatric lineages. Several recent studies suggest such patterns are the norm among coral species. For instance, along the Florida reef tract, both *Siderastrea siderea* and *Montastraea cavernosa* show similar population structure to what we observe: highly diverged lineages, sometimes occurring sympatrically, and which differ across habitat types (in their case depth)[45]. In Panama, *Orbicella faveolata* populations harbor various distinct lineages, and these differ in thermal tolerance[59]. In Florida, *Porites astreoides* contains sympatric lineages that differ in thermal tolerance under experimental stress[7]. *Acropora hyacinthus* forms several strongly differentiated genetic lineages across mainland Japan and the Ryukyus archipelago, with one lineage appearing more adapted to colder temperatures and dominating the species' poleward range expansion in that region[60]. This recent work, along with our findings, suggests that reef-building corals specialize to occupy narrow environmental niches, generating strong genetic differentiation across small spatial scales and even in sympatry. The high divergence ($F_{ST}$) we observed between lineages in our RAD analysis suggest cryptic speciation in *P. lobata*. These values could be inflated by technical

artifacts (i.e., PCR duplicates), although a recent study showed high (95%) agreement in RAD genotypes before and after removal of PCR duplicates[61]. Whether these patterns may represent speciation in progress or simply be a characteristic of coral genetic diversity remains to be resolved and has important implications in the context of future species conservation efforts. Future studies should test for cross-fertilization compatibility and post-settlement survival between coral lineages directly to resolve this question.

A central question within coral biology and conservation efforts is whether there are trade-offs between thermal tolerance and other key traits like growth or fecundity[48]. In addition to warm temperatures, the Rock Islands have pH and aragonite saturation levels near those expected in the open ocean in 2100[14,15]. Low pH and aragonite saturation can compromise coral growth and especially impacts skeletal density and facilitates bioerosion[42,43]. While other environmental conditions in the Rock Islands, such as light levels or turbidity, can also influence growth[62], it is still worth examining potential growth and thermal tolerance trade-offs in the Rock Islands given their extreme pH and aragonite saturation conditions. LB corals were able to maintain high thermal tolerance and consistent growth regardless of pH conditions, indicating this lineage does not show strong trade-offs in its ability to handle multiple stressors (Fig. 5). In contrast, the outer reef-associated DB corals grew less and had lower density calcification rates when found in the Rock Islands, where they show a trend toward higher thermal tolerance (Fig. 5). Though one could interpret the DB's lower growth as a trade-off with thermal tolerance, the challenging conditions for calcification in the Rock Islands are more likely to be driving factors, especially since this lineage shows low thermal tolerance overall. The RD lineage, which is nearly exclusively found in the two lowest pH sites, Risong and Mercherchar[14,15], shows lower growth metrics than the LB lineage, suggesting Rock Island conditions do have the potential to hinder coral growth (Figs. 1 and 3). Without being able to compare growth of the RD lineage under more favorable pH conditions, however, it is not possible to evaluate any trade-offs between its high thermal tolerance and growth (there was only one RD coral sampled in an outer reef and it was not cored). It appears that any combinations of the environmental factors across Rock Island and outer reefs sites do not affect the LB lineage in a substantial way, as it is able to maintain both growth and thermal tolerance across all habitats (Fig. 5). Thus, whatever trade-offs may exist, they do not appear to be ubiquitous. It is unclear why favorable traits associated with the LB lineage would not become fixed in Palauan populations of *P.* cf. *lobata*, but there could conceivably be trade-offs in coral physiology that we did not measure[63].

The thermal tolerance of LB corals and their consistent growth across habitats have important implications for understanding the potential of corals to withstand future environmental changes. Many reef systems are characterized by variable or warmer thermal regimes across space and through time, regimes that can select for and harbor thermally tolerant genotypes. Additionally, thermal tolerance appears to be highly heritable in reef-building corals[47,64–67]. Our results demonstrate that warmer environments can serve as breeding grounds for more tolerant corals (e.g., the LB and RD lineages) and that some of these (e.g., the LB lineage) can (1) thrive and maintain their tolerance even when they disperse to cooler environments and (2) do so without growth obvious trade-offs. Based on its high thermal tolerance and apparent lack of trade-offs, the LB lineage may be particularly well-suited to survive through subsequent increases in temperature and declining pH. Gaining a better understanding of the mechanisms behind its resilience can also provide valuable insights into coral physiology and the future of coral reef ecosystems.

As oceans worldwide continue to warm, corals derived from extreme habitats will be at a competitive advantage and may enable the survival of otherwise vulnerable reefs. Identifying and safeguarding natural breeding grounds of environmentally tolerant corals that can thrive under future climate conditions will be fundamental to the persistence of coral reef ecosystems worldwide in the coming decades. Nevertheless, the reality remains that curtailing climate change and the greenhouse gas emissions that cause it will be the only way to truly safeguard our planet's biodiversity.

## Methods

**Coral sampling**. Between 2011–2018, we collected tissue from 543 *Porites* cf. *lobata* colonies using a hammer and chisel while on SCUBA (Supplementary Data 2). Tissue was preserved in RNAlater™ (Invitrogen, Waltham, MA), incubated overnight at 4 °C, and frozen at −20 °C ($N = 329$), or frozen directly at −80 °C ($N = 20$), or preserved in 95% ethanol and frozen at −20 °C ($N = 194$) until DNA extraction. Colonies were sampled haphazardly within each site, across 13 sites (Fig. 2), based on morphological characteristics of *Porites lobata* detailed in ref. [68].

The site "Ngermid" has been referred to as "Nikko Bay" in previous publications. We use "Ngermid" here as that is the name preferred by Palauan natives. The "Outer Taoch" site contained samples from three outer reef locations: Airai (GPS coordinates: 7.33210, 134.56020, $N = 5$), Rael Dil (7.24990, 134.45073, $N = 3$), and a fringing reef (7.27193, 134.38115, $N = 30$) immediately outside of Taoch Bay. The coordinates for this last site were used for mapping because most of the samples in this group are from this location. Due to the presence of various lineages within our dataset, population genetic metrics (e.g., $F_{ST}$) were not calculated by collection site, so this choice should have no bearing on results.

**DNA extraction and polymerase chain reaction (PCR) of microsatellites**. Samples were processed as per[20]. Briefly, a thawed ~1 mm² piece of coral was homogenized into a fine powder using a new standard safety razor blade sterilized with ethyl alcohol and flamed. The homogenate was processed using the Qiagen® DNeasy Blood and Tissue DNA extraction kit according to the manufacturer instructions, with a modified Proteinase K incubation of at least 24 h. Negative controls ($N = 5$) without any coral tissue added were included every 70 samples and subjected to all the same downstream processing and analyses.

We amplified 14 microsatellite markers with fluorescently labeled primers developed for *P.* cf. *lobata* by[20,69]. PCR settings were: (1) initial denaturation at 94 °C for 5 min, (2) 35 cycles of [94 °C for 20 s, annealing at 52, 54, or 56 °C (plex-dependent) for 20 s, 72 °C for 30 s], and (3) final extension for 30 min at 72 °C. The Pennsylvania State University Nucleic Acid Facility measured fragments on an ABI 3730 (GeneScan) with a LIZ-500 internal size standard.

**Microsatellite multi-locus genotyping**. We used GENEMAPPER™ v3.0 (Applied Biosystems) to visualize electropherograms and call alleles. Scoring was conducted blind to the site of origin for each sample. The first author scored alleles three separate times from scratch for all samples. Downstream analyses and results were consistent for all three sets. All automated allele calls were verified and curated manually to ensure accuracy and consistency between samples. After initial manual verification of all samples, raw allele sizes and allele call designations were exported and explored graphically. Boxplots of allele sizes by allele call designations were plotted for all markers, and all data points outside of the interquartile range were re-verified manually and removed if peaks were of poor quality (i.e., very low height, non-standard shape, or possibly a spectral pull-up artifact from another channel that was not automatically detected). We also plotted allele size density curves by allele call designations, to identify samples with similar allele sizes but called as separate alleles. These techniques were used over several iterations to ensure allele calls were clean and of high quality.

For samples showing more than two alleles at a locus, the following steps were taken to select two alleles for population genetic analysis:

(1) For samples that were run more than once for any marker and showed a third allele in only one run, the singleton allele was dropped.
(2) For samples run only once for a marker, the third allele was dropped if its height was less than half the second highest peak of the other two alleles.
(3) For samples that were run more than once and showed 3 or more alleles consistently, or the sample was run only once but all alleles had roughly equal peak heights, the two alleles with the higher frequencies in the whole dataset were retained (i.e., the rarest allele(s) were dropped). This choice was made because it would be less likely to bias downstream analyses toward isolated populations.

Two microsatellite markers were dropped due to high rates of missing data (>40%). Samples with fewer than 10 of the remaining 12 loci were excluded ($N = 26$); 322 samples were retained.

**RAD-sequencing library preparation and sequencing**. A subset of samples with sufficient quality extracted DNA were then processed for RAD-sequencing. Genomic DNA concentrations were standardized using a Qubit™ 2.0 fluorometer (Invitrogen) to 20 ng/μl. A total of 50 μl per sample was sent to Floragenex (Portland, Oregon) for single enzyme RAD library preparation with PstI enzyme digestion. Eighteen PCR cycles were used for library amplification. Each sample was identified by a unique 10 nucleotide barcode. Samples ($N = 185$) were sequenced as 100 base pair single end reads across 6 lanes of an Illumina Hiseq 4000™ using v4 chemistry at the University of Oregon Genomics Core facility.

**RAD-seq processing and SNP-calling**. Raw reads were processed using the "process_radtags" module of Stacks v.1.46[70], allowing for up to three mismatches in the sample barcode (this was the maximum number of mismatches at which the barcodes remained unique). Reads with low quality scores (PHRED <10) across a sliding window of 15% of the read length were discarded. We retained 78% of the original reads. Average sequencing depth was 8.5 million reads per sample. Three samples replicated within the plate showed 1.5–2-fold variability in sequencing depth. One sample which had an unusually high number of reads (>35 million) was discarded.

For read mapping and SNP calling, we used the dDocent v.2.3.7 pipeline[71] with a *Porites lutea* or *P. lutea* symbiont draft genome obtained from the REFUGE 2020 database (http://refuge2020.reefgenomics.org/) as reference (for coral and symbionts SNPs, respectively). dDocent clustered reads based on >95% similarity using CD-HIT v.4.6.8[72], mapped reads to the reference using the MEM algorithm of BWA v.0.7.17[73] with a match score of 1, mismatch score of 3, and gap-open penalty of 4, and called SNPs using FreeBayes v.1.1.0[74] with default values (E = 3, m = PHRED 10, q = PHRED10, -V, and using the sampling sites as the population designations). The resulting "TotalRawSNPs.vcf" file was filtered using vcftools v.0.1.15[75] and vcffilter (https://github.com/jameshicks/vcffilter) following the suggestions in the dDocent manual, with a final thinning (-thin option in vcftools to keep only SNPs more than 150 bp apart, e.g., only one SNP per rad tag) to obtain a final set of 12,761 bi-allelic SNPs in 146 retained samples for the coral host and 218 bi-allelic SNPs in 137 samples for the symbiont.

**Population structure**. Coral host SNPs and microsatellite data were run through STRUCTURE v.2.3.4[54] using an admixture model with correlated allele frequencies and default parameters following previously used settings for corals[19,20]. Including sampling (geographic) information in the prior did not affect results and is not reported. MCMC chain settings were: $1 \times 10^5$ burn-in, $1 \times 10^6$ iterations from $K = 1$ to $K = 12$, with ten replicate chains per $K$. We used the CLUMPAK feature[76] on the STRUCTURE Selector webserver to combine and visualize STRUCTURE output through the "main pipeline" option with CLUMPP parameters: LargeK-Greedy search, 10,000 random input orders, dynamic MCL, and default minimal cluster size. The Structure Selector webserver was used to run "Best K" metrics which included methods to evaluate the optimal $K$ described in refs.[77–79]. Coral samples were assigned to their majority group (>50% ancestry for $K = 4$) as their "final" lineage for further analyses, following convention in other studies[60]. In the three cases where the majority lineage differed between microsatellite STRUCTURE and RAD-seq assignments, the RAD-seq assignment was used (Fig. S2). It should be noted that only one of the samples with disagreeing assignments had core data.

The R package adegenet[80] was used to explore genetic structure in both microsatellite and RAD-seq data for both coral and symbionts. Principal component analyses were conducted using the dudi.pca() function on scaled and centered genind/genlight objects. We used the find.clusters() functions to select the optimal number of groups based on Bayesian Information Criterion (BIC). Discriminant analyses of principal component (DAPC) was used to visualize clusters, with the number of principal components retained determined through cross-validation xdapval() to avoid overfitting. For symbiont RAD-seq data, PCAs and DAPC were colored based on coral host lineage assignments (though these also perfectly matched the identified clusters through DAPC). Nei's FST between lineages was calculated using the stamppFst() function from the R package STaMPP for RAD-seq data, using 100 bootstraps for estimating significance. For microsatellite data, we used the function genet.dist() function from the package hierfstat[81].

**Temperature records and analyses**. The Coral Reef Research Foundation (http://wtc.coralreefpalau.org/) provided 30-min interval in situ temperature data for Mecherchar, Helen, Drop Off, Ngerdiluches, Ngerchelong, Ngermid, and Kayangel. These data were recorded using U22 loggers (Onset Technologies, MA). Temperature data covered periods from 2010–2017 and from 2–15 meters depth. The foundation indicated accuracy was determined to be within 0.1 °C through pre- and post-deployment calibration against a NIST traceable mercury thermometer and that individual thermographs were also cross calibrated with each other. Temperatures for Risong and Taoch were obtained from U22 loggers (Onset Technologies, MA) deployed between 2–5 meters depth by the Cohen Lab at Woods Hole Oceanographic Institution from 2011–2013 recording at 15-min intervals.

Statistical metrics of temperature time series for each site were calculated using the "zoo"[82] and "xts"[83] packages in R. To examine daily temperature patterns, each

time series was filtered in MATLAB 2015a using a bandpass Butterworth filter to retain signals between 5 and 30 h in frequency and remove seasonal fluctuations. Daily range (maximum-minimum temperatures) were calculated for each site in R.

**Coral core sampling and analysis**. Coral cores ($N = 121$) were taken using an underwater pneumatic drill equipped with a diamond-tipped drill bit powered by compressed air from a SCUBA tank. Cores ranged from 10 to 204 cm long, covering years from 1970 to 2014. Cores were dried in an oven and imaged using a Volume Zoom Helical Computerized Tomography (CT) Scanner at Woods Hole Oceanographic Institution. Scans were analyzed using an automated computer program developed and described in ref. [43] and modified by ref. [30]. Cores which displayed high levels of bioerosion, inconsistent banding patterns, partial mortality or other deformities that did not allow for accurate identification of annual growth bands and, as a result, correct identification of years corresponding to bleaching events, were excluded from analysis ($N = 42$). Of the remaining 80 cores, 11 had unsuccessful DNA extractions or failed other downstream genetic data filters, leaving 69 corals for final analyses.

While differences in growth patterns and stress band prevalence between RI and OR habitats had been previously shown[12], we wished to test if genetic population groups provided additional explanation of these responses. A colony's genetic group was assigned as its predominant (>50%) STRUCTURE assigned group for $K = 4$, which was the best $K$ across several methods. The growth patterns and presence/absence of stress bands during the 1998 and 2010 bleaching events ($N = 44$, 5 sites), are data previously described and published in refs. [12,43]. An additional 14 cores were analyzed for this study.

Growth metrics (extension and calcification rates and density) were calculated for each core by averaging the yearly growth during the years 1999–2009, in order to avoid bleaching years that might influence growth patterns. Differences among lineages in growth parameters were tested using ANOVA with post hoc Tukey tests.

Stress bands were defined as a region of the core at least 1 mm thick in which density and the change in density gradient exceeded two standard deviations above each individual whole core average density and gradient, following the definition and methods in refs. [12,43]. Differences in the proportion of cores showing stress bands between lineages was tested using a Chi-Squared test.

**Symbiodiniceae inter-transcribed spacer-2 (ITS2) denaturing gradient gel electrophoresis (DGGE) genotyping and sequencing**. We analyzed 27 coral samples representing all four lineages, across 8 outer reef and Rock Island sites. To test whether the dominant symbiont community of coral colonies shifted across Palau's strong environmental gradients, we amplified the ITS2 region of Symbiodiniceae's nuclear ribosomal DNA and visualized bands using DGGE following protocols in refs. [84,85]. Briefly, the "ITSintfor2" and "ITS2clamp" primers were used for initial amplification with a touchdown PCR protocol consisting of: (1) initial denaturation at 94 °C for 2 min, (2) 20 cycles of [94 °C for 20 s, initial annealing temperature of 62 °C for 10 s and decreasing at 0.5 °C intervals every cycle until 52 °C, then 68 °C for 30 s], (3) continuing with another 18 cycles at annealing temperature of 52 °C, and (4) a final extension for 10 min at 68 °C.

Products were loaded onto 8% acrylamide gels with a 40–75% denaturing gradient and run for 24 h at 90 volts. The gels were stained in 1 liter of deionized water with 10 μl of SYBR Red™ (Thermo Fisher) for 30 min, de-stained in 1 liter of deionized water for 30 min, and then visualized with a UV gel imager. DNA from *Cladocopium* (C15) cultures was obtained from the LaJeunesse Laboratory (Pennsylvania State University) and run alongside Palauan samples for band identification.

Representatives of any additional bands seen in Palauan samples were excised using a sterile pipette tip, homogenized in 5 μl of molecular grade water, and reamplified using the "ITS2intfor2" and "ITS2rev" primers and a standard PCR protocol: (1) initial denaturation at 92 °C for 3 min, (2) 35 cycles of [92 °C for 30 s, annealing at 52 °C for 40 s, 72 °C for 30 s], and (3) final extension for 10 min at 72 °C. Products were visualized on a 1% agarose TAE gel, and successfully re-amplified samples were purified using a MinElute™ PCR Cleanup Kit (Qiagen) and sent for Sanger sequencing at Sequegen (Worcester, MA). Sequences were then aligned to Symbiodiniceae sequences on the NCBI "nt" database using the MEGABLAST algorithm with default parameters on the NCBI BLAST webserver.

**Statistics and reproducibility**. The statistical tests used for analyses are described for each type of data above, along with the corresponding R or other software packages that were used to obtain results and their versions. Scripts for such analyses are also available (see Data availability statement below). The number of replicates in each analysis is listed throughout the manuscript when the results are presented. The reader should note that the number of corals with cores is much smaller than the number of corals sampled for genetic analyses. This information is all detailed in "Coral sore sampling and analysis" section above. Supplementary Data 2 contains a list of all the samples processed and their corresponding collection metadata. Core-derived growth data are available in Supplementary Data 3, and core stress band data in Supplementary Data 4. Supplementary Data 3 and 4 are also available in the github repository as these are used as input files for analyses (named core_data.txt and strata_all_samples.txt on github, respectively).

**Reporting summary**. Further information on research design is available in the Nature Portfolio Reporting Summary linked to this article.

## Data availability

All data associated with this manuscript are available in the Supplementary Information (Supplementary Data 2–4) or appropriate databases: RAD-sequencing data are available on NCBI's SRA under accession number PRJNA801929. Additional input data are available in the github repository of H.E.R. at https://github.com/hrivera28/Palau_porites. The only exception (due to large memory requirements) is for raw coral core CAT scan files, which are available upon request to A.L.C. Coral samples were imported to the U.S. under CITES permit numbers: PW14-163, PW15-022, PW18-121.

## Code availability

All scripts used to analyze the data presented here are available in the github repository of H.E.R. at https://github.com/hrivera28/Palau_porites. Versions of all R packages used are listed at the end of each script.

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

## Acknowledgements

First, we extend our sincerest gratitude to the Palau International Coral Reef Center (PICRC) as well as Palauan government for permission to conduct this work, including the states of Hatohobei, Koror, and Kayangel. We thank Yimnang Golbuu, Marine Gouezo, Joy Schmull, and Geraldine Rengiil of PICRC for assistance with permitting and sampling logistics. We thank Hannah Barkley for feedback on this manuscript, fieldwork assistance, and invaluable mentorship to HER. We also thank Kathryn Rose-Pietro, Pat Lohmann, Tom De Carlo, and the crew of R/V *Alucia* for fieldwork and sampling assistance, Timothy Shank for use of his thermocycler, Meghann Devlin-Durante and Jennifer Boulay for training in microsatellite analyses, and Ellie Bors for assistance with RAD techniques. We thank Ann Tarrant for laboratory space and supplies and comments on earlier versions of this manuscript, Patrick Colin from the Coral Reef Research Foundation for in situ temperature data, and Andy Solow and Vicke Starczek for guidance on statistical analyses. We also thank Carolyn Tepolt for suggestions on population genetics analyses, and Simon Thorrold, Annick Cros, and Sarah Davies for comments on earlier versions of this manuscript. Lastly, we acknowledge the following funding sources for making this work possible: To A.L.C.: National Science Foundation (OCE-2049567), The Seija Family, The Arthur Vining Davis Foundation, the Atlantic Charter Donor Advised Fund, and the Dalio Foundation, Inc. To J.R.T.: The Singapore Ministry of Education and National Research Foundation through an RCE award to Singapore Centre for Environmental Life Sciences Engineering (SCELSE), and the MIT Sea Grant Office. To H.E.R.: Woods Hole Oceanographic Institution Coastal Ocean Institute Grant and Ocean Venture Fund, National Defense Science and Engineering Graduate Fellowship Program, the Martin Family Fellowship for Sustainability, and the American Association of University Women Dissertation Fellowship. To M.D.F.: The WHOI Postdoctoral Fellowship. To K.S.M.-K. and H.E.R.: Paul M. Angell Family Foundation Grant. To I.B.B.: OCE-1537959.

## Author contributions

Conceptualization, methodology: H.E.R., A.L.C., and K.S.M.-K. Writing—review and editing: Lead: H.E.R., K.S.M.-K., and A.L.C., Supporting: J.R.T., I.B.B., and M.D.F. Formal analysis and investigation: H.E.R. and K.S.M.-K. Data curation, visualization, and writing—original draft preparation: Lead: H.E.R, Supporting: A.L.C., J.R.T., K.S.M.-K., M.D.F., and I.B.B. Funding acquisition, project administration, and resources: Lead: A.L.C. and K.S.M.-K., Supporting: H.E.R., J.R.T., and I.B.B.

## Competing interests

The authors declare no competing interests.
