## [Peer Review File · Communications Biology]

Reviewers' comments:

Reviewer #1 (Remarks to the Author):

This manuscript presents the findings of Rivera and colleagues who studied bleaching patterns in *Porites* cf. *lobata* across large environmental gradients in Palau. The authors found that *Porites* cf. *lobata* is comprised of at least 4 'lineages' whose abundances are unequal across geographic sites, with some being more common in Palau's rock islands while others are more common in outer reef environments. These lineages are also biologically differentiated in terms of their calcification rate, extension rate and stress band prevalence (a proxy of bleaching stress), largely driven by the 'RD' lineage. Overall this paper is a solid contribution to the field, in particular the joint analysis of growth rates with bleaching stress and the consideration of adaptation to smaller-scale environmental variation within an island archipelago, which is particularly lacking in marine studies. The statistical tests were appropriate, although I would suggest the authors perform additional analyses to identify patterns of local adaptation within each lineage, which would complement their approach describing differences between the lineages. The authors could strengthen the manuscript by more carefully considering their results across different levels of biological organization. My impression is that the authors have chosen the word 'lineage' to avoid wading into a discussion on cryptic species versus immense levels of population genetic structure, but in the process of trying to remain agnostic the analysis and discussion have lost a great deal of clarity. Furthermore, a more careful consideration of the role of hosts and symbionts in this system would greatly improve the manuscript, as this species is one where host-symbiont coevolution and local adaptation in either partner could dramatically alter the bleaching response of the holobiont.

Major Concerns

The authors have chosen to remain agnostic about whether these lineages represent cryptic species because they were unable to test the biological species concept (l. 144-147). This choice is the most conservative, it's true that there have been no tests of successful matings between the lineages, the gold standard for defining species on the criteria of reproductive isolation. However, the genetic patterns that the authors present in this study beg the question of what is maintaining very high levels of F_{ST} (~0.7!) in sympatric populations. Unfortunately, much of the discussion hinges on this designation and its ambiguity leads to a lack of clarity about the ecological and evolutionary consequences of bleaching resistance, local adaptation and growth tradeoffs in this (or these) species. For example, beginning on l. 221 "Limited connectivity between the Rock Island and outer reefs as suggested by hydrodynamic model output likely facilitates local adaptation, which is corroborated by the population differentiation seen here." Local adaptation is a population-level phenomenon, and it is therefore important to identify and characterize what constitutes a population. If the lineages represent subpopulations, then this is a reasonable interpretation, but the authors should consider what is maintaining such high levels of genetic differentiation in the Discussion. If the lineages are distinct species, then local adaptation will occur in subpopulations within each lineage, and statistical tests of this hypothesis should be carried out on individuals from the same lineage living in different habitats.

The discussion of locally adapted lineages would also be improved by more carefully describing the level (intra- vs. inter- lineage) of variation that is being discussed. On line 208, for example, "the Rock Island-associated LB lineage maintains higher thermal tolerance across habitats, without showing any corresponding trade-offs in growth metrics, indicating this lineage may be particularly well suited for restoration" requires an explanation of why this lineage hasn't become dominant in all habitats if it grows quickly, doesn't bleach and experiences no tradeoffs. If the lineages are subpopulations, the question is why these traits don't sweep to fixation. If the lineages are different species, why does one not competitively exclude the others? In some ways these are similar questions, but their distinction matters from an ecological and evolutionary standpoint, and will have more immediate consequences for conservation and management decisions. If lineages do not interbreed, for example, then intervention strategies that attempt to breed the traits of LB into another lineage will not be possible.

Additionally, the authors point out that the LB lineage might be good candidate for restoration (l. 310), but this ignores the potential ecological significance of RD, DB and PI, especially if they occupy different environmental niches, as discussed on l. 280-281. Furthermore, if the lineages cannot interbreed, local adaptation would occur independently within each lineage, and the contrasts to which the authors draw the reader's attention (Figs. 2-3 showing differentiation between the lineages; l.223) are unsuitable. Instead, Fig. 5 would be a better example of patterns consistent with local adaptation to the Rock Island vs. Outer Reef habitats within each lineage (but not sufficient by themselves without experimental follow-up) if each lineage is independent and has subpopulations in these two habitats. One analysis that would add evidence for within-lineage local adaptation is to identify population structure in the lineages themselves between the Rock Island and outer reef sites. This would reveal patterns consistent with within-lineage local adaptation between populations inhabiting different environments, for instance population structure between Rock Island and outer reef sites in LB and DB.

Growth and bleaching are likely impacted by many underlying genetic mechanisms as well as environmental mechanisms. The authors point this out in *A. hyacinthus* in American Samoa, but should consider contextualizing their results in light of recent work highlighting a similar pattern of bleaching resistance associated with increased reef temperatures in *A. hyacinthus* in Palau (Cornwell et al. 2021). If 'lineages' are cryptic species, it could be that each lineage has to contend with a different set of mechanisms (invisible to us until we figure out what to measure) that might lead to trade-offs, but that might only be evident when examining each lineage by itself. Emerging lines of evidence for tradeoffs are now in the literature, including a growth/symbiont density trade-off (Cornwell et al. 2021) that also might impact bleaching severity (Cunning and Baker 2013). On the other hand, the discussion should have a completely different focus if these lineages represent individuals that are all from the same species, instead of cryptic species. What exactly is keeping them so genetically differentiated? Given that the SNPs are from a RAD dataset, one would expect most of them to be selectively neutral, the same goes for the microsatellites. It is difficult to come up with an explanation for what would maintain 'lineages' that are identified using neutral loci without invoking some sort of between-lineage reproductive failure (either pre- or post-zygotic). The authors advance the possibility that there are hybrids (l. 143), an interesting hypothesis to be sure, and one that the authors might be able to bolster using other analytical techniques (e.g. F4 statistics; Reich et al. 2009 Treemix; Pickrell and Pritchard 2012). It's worth mentioning that the lineage with very high levels of FST (PI) only looks admixed in one individual, and that individual has ancestry from all 4 lineages (Fig. 3 11th bar from the left in the Risong population) which suggests ambiguity in its assignment as a possible alternate hypothesis to an admixed origin.

Finally, the contribution of the symbionts to the patterns described in the study deserves more discussion, especially given that this coral species transmits them vertically from mother to offspring. Vertical transmission is a very effective way of aligning the fitness interests of both the hosts and symbionts, which means that local adaptation in the symbionts can be highly beneficial to the host in the current generation as well as future generations where host-symbiont lineage combinations will remain intact. Treating the symbionts as a homogeneous population devoid of variation is inappropriate on the basis of a single genetic locus. At best it ignores neutral variation, at worst the symbionts are major drivers of these patterns, but that variation remains hidden. It is challenging to generate multilocus genomic datasets for the symbionts, but recent studies suggest that they are highly genetically structured by geography (Thornhill et al. 2017; Cooke et al. 2020; Kriefall et al. 2022), host lineage (Forsman et al. 2020); possibly by environment (Klueter et al. 2015; Cornwell and Hernandez 2021; Kriefall et al. 2022). Many (but not all) of these patterns would be largely undetectable at the ITS locus, and the discussion should consider this when interpreting phenotypic variation and patterns of local adaptation in the coral 'lineages.'

Minor Considerations:

The authors should consider explicitly stating that symbionts in this species are vertically transmitted in the introduction.

I. 183 change 'confined two' to 'confined to two'

I. 186 – Why only DB/LB and not PI? Seems like PI has sample sizes in the Rock Islands approaching the sample size of DB in the same habitat.

I. 186-198 – The authors should consider reframing this paragraph into two sets of comparisons. First, within-lineage comparisons (e.g. LB in rock islands compared to LB on outer reef). Second, between-lineage comparisons. Are there differences between LB and DB in both places? These two sets of comparisons would set up an analysis for examining local adaptation (see Kawecki and Ebert 2004 for a well-reasoned discussion of how to approach these analyses). This is important, it looks like LB performs the same or worse than DB on outer reefs which points to potential tradeoffs that the authors argue against later in the text (I. 293). If there are no tradeoffs the statistical comparisons that inform that conclusion should be clearer, and I would suggest a power analysis to determine what effect size could have been detected with these sample sizes.

I. 204 – Rephrase 'promoted the development of thermally tolerant lineages.' Do you mean that the Rock Islands create an environment that selects for thermally tolerant lineages? Aside from a lack of clarity it also makes it sound like the Rock Islands are the reason these lineages are physiologically and genetically different, but there is no evidence for that in this study.

I. 208 – Specify if 'driven by genetic factors' means within the host, symbiont or both. See above for why it could just as easily be the symbionts given that they are vertically transmitted.

I. 217-219 – Is it possible that you have more evidence for this in the core data? Do those samples allow you to see that colonies settled after the last heat wave? It could be an interesting addition if individuals from less stress tolerant lineages are all young and settled after the last heating event, the hypothesis being that they have yet to experience a strong selective filter.

I. 221-223 – Can't your genetic data shed some light on whether there are high levels of gene flow between the rock islands and the outer reefs? If the Rock Islands are selecting for only thermally tolerant individuals you would expect higher rates of gene flow between rock island sites than between rock island and outer reef sites within each lineage.

I. 230 – 'produce' should be replaced, do you mean 'select for' or are you advocating that the Rock Islands are the actual location where these thermally tolerant lineages originated? I'd believe the former but not necessarily the latter (absent much more data from across the geographic range of the species).

I. 274 – Is it fair to consider diverged lineages that occur at different depths 'sympatric'?

I. 236 – See comment for I. 214-215, the comparison to American Samoa *A. hyacinthus* populations is apt but comparing to Palau populations seems even more relevant.

I. 305-307 – Consider adding a discussion of what the appropriate spatial scale is to be asking about local adaptation in these populations. Outer reef and Rock Islands habitats are clearly different, but are there microhabitats in each that provide refuge for thermally (in)tolerant lineages? One reason why LB looks unaffected across habitats could be that it is good at finding ideal microhabitats in each location.

I. 321 – Rephrase 'increase selective pressures.' Do you mean that long residence times ensure that larvae from each generation tend to land in the same geographic location as their parents, thus exposing them to consistent selection pressures across generations?

I. 321-323 – Without more population genetic data this really seems like a stretch. If you are going to

make the argument that they originated in the rock islands you should show more data to bolster it. There is no evidence presented here that the Rock Island corals are a closed population, and have been that way for 30-50 generations.

I. 403 – RAD data are from single end reads, does this mean that the authors could not detect and remove PCR duplicates arising during library construction? It leads me to wonder if this is partly driving the exceedingly high levels of FST between the lineages. The agreement between the RAD data set and microsatellites implies that it isn't playing such a large role in identifying members of each 'lineage' but perhaps is inflating estimates of differentiation if heterozygotes are missed. Or was there some sort of random identifier in the barcodes that allowed the authors to overcome this problem?

I. 431 – remove 'using'

Fig. 1B – Change 'Nikko' to 'Ngermid' to stay consistent with I. 343

Fig. 5 – Can you use the same visual language of '*' to show the statistical comparisons you made and whether they were significant as in Fig. 4? This is especially important for making an argument about local adaptation within each lineage. Also, to reiterate a comment from above, why not include PI in these plots?

I. 611 – Consider changing 'colored by their dominant' to 'colored by their dominant lineage'

I. 764 and 766 – Are there peer reviewed versions of these publications?

Cooke et al. Genomic signatures in the coral holobiont reveal host adaptations driven by Holocene climate change and reef specific symbionts (2020). *Science Advances* 6:eabs6318.

Cornwell et al. Widespread variation in heat tolerance and symbiont load are associated with growth tradeoffs in the coral *Acropora hyacinthus* in Palau (2021). *eLife* 2021;10:e64790

Cornwell, B. and Hernández, L. (2021). Genetic structure in the endosymbiont *Breviolum 'muscatinei'* is

correlated with geographical location, environment and host species. *Proc. R. Soc.*

B.2882020289620202896

Cunning, R., Baker, A. Excess algal symbionts increase the susceptibility of reef corals to bleaching. *Nature Clim Change* 3, 259–262 (2013). <https://doi.org/10.1038/nclimate1711>

Forsman, Z.H., Ritson-Williams, R., Tisthammer, K. et al. Host-symbiont coevolution, cryptic structure,

and bleaching susceptibility, in a coral species complex (Scleractinia; Poritidae). *Sci Rep* 10, 16995 (2020). <https://doi.org/10.1038/s41598-020-73501-6>

Kawecki, T.J. and Ebert, D. (2004), Conceptual issues in local adaptation. *Ecology Letters*, 7: 1225-1241.

<https://doi.org/10.1111/j.1461-0248.2004.00684.x>

Klueter, A.; Crandall, J.B.; Archer, F.I.; Teece, M.A.; Coffroth, M.A. Taxonomic and Environmental Variation of Metabolite Profiles in Marine Dinoflagellates of the Genus *Symbiodinium*. *Metabolites* 2015, 5, 74-99. <https://doi.org/10.3390/metabo5010074>

Kriefall N. G., Kanke M. R., Aglyamova G. V. and Davies S. W. 2022. Reef environments shape microbial

partners in a highly connected coral population. *Proc. R. Soc. B*.2892021245920212459

<http://doi.org/10.1098/rspb.2021.2459>

Pickrell JK, Pritchard JK (2012) Inference of population splits and mixtures from genome-wide allele frequency data. *PLoS Genetics* 8, e1002967.

Reich D, Thangaraj K, Patterson N, Price AL, Singh L (2009) Reconstructing Indian population history. *Nature* 461, 489-494.

Thornhill, D.J., Howells, E.J., Wham, D.C., Steury, T.D. and Santos, S.R. (2017), Population genetics

of
reef coral endosymbionts (Symbiodinium, Dinophyceae). *Mol Ecol*, 26: 2640-2659.
<https://doi.org/10.1111/mec.14055>

Reviewer #2 (Remarks to the Author):

Overall

This is an important study that falls short in the presentation of the data. It feels as though the authors are trying to pull more from their paper than what the data supports. This is unfortunate because the paper stands on its own, but the authors need to reign in (or better present) some of their claim, as well as provide more information on growth analyses.

First, the authors need to include discussion about the drawbacks of using massive *Porites* as their organism for study. This group of species are naturally very stress tolerant. In fact, they tend to be one of the few species that is always found in very marginal environments. Massive *Porites* aren't building reefs like acroporids, except in the most extreme of environmental conditions that limit the presence of all other coral species. Furthermore, this species possesses a very porous skeleton that allows the coral to retract its tissues deep into the skeleton during times of stress, including bleaching. The discussion needs to be written so that the authors acknowledge and defend using a species of coral that will likely see the smallest population decline of any coral on the planet with global warming.

Second, the discussion of temperatures and thermal stress between the rock islands and offshore sites is not clear. Phrases like thermal anomaly and heat stress are used without any quantification of what the thermal anomaly or thermal stress actually is. The absolute temperatures, while interesting and show us that the corals persist in a warmer environment, tell us nothing about thermal stress as it pertains to the anomaly from a climatology, or average conditions. Just because it's hotter in the Rock Islands does not mean that these sites experience more thermal stress, nor does it mean that these sites experienced the same levels of thermal stress and bleaching that was reported on the outer reefs in the past. What are the climatologies and thermal anomalies for the offshore versus inshore reefs? What about the history of thermal stress for these two locations. This is not discussed, nor expounded on. It is the thermal anomaly, or deviation from the climatology or long-term averages, that matters when it comes to bleaching.

Third, density bands are not fully accepted by the sclerochronology community as a means to ID past bleaching events. I realize there was a published study – but this was from an ENSO dominated location with huge thermal anomalies that is an outlier to what most reefs experience. You can't use density bands as the only evidence for past thermal stress and bleaching.

Finally, the growth data are presented in a disingenuous manner and the data have been massaged to serve the narrative that there aren't growth trade-offs. The authors are reaching for a result that their data do not support. By looking at Figure 4, I'd say there most likely ARE growth trade-offs to living in very hot environments. Lineage RD, which only occurs in the hottest and most thermally variable (i.e., most extreme) environments of Risong and Mecherchar, clearly has very low density, calcification and extension. For the lineage that co-exists across offshore and Rock Islands (e.g., LB), this is only found in the middle of the channel, where there is much more water exchange and not where the environment is most extreme. In other words, they exist somewhere in between the stable, cool offshore conditions and the most hot and variable environments, which restricted lineages.

The issue with the core data is that there is zero discussion about how the data were averaged. The authors say they use prior and new core data, but what years do these cover? Did all your cores cover the exact same number of years? If not, the data may be biased. Did you include both known bleaching and non-bleaching years in your averaging of growth rates? What about number of data

points per year per site? I'm guessing the cores at these inshore sites had a lot of alteration, including bioerosion, and that getting data from them may have been more challenging than the authors indicate. The second author on this paper is a world leader in coral core analysis and the way these data are handled and reported is very disappointing. This is a shame because the paper does not hinge on these data at all and stands on its own without needing to cryptically package the growth data to try and claim there is no growth tradeoff.

Specific comments

- 1)Line 29 – is there evidence that the rock islands experience more extreme heatwaves? Where is the proof?
- 2)Line 62-63: Historically? Or is this post-1997/98 bleaching? Be specific as this matters greatly. I'm guessing species richness was historically less in the Rock Islands, but I could be wrong
- 3)Line 66: Are there really greater anomalies or is the absolute temperature just higher? What are the anomalies? What is the climatology for each site?
- 4)Line 87-89: Again, it's the temperature anomaly that matters with regards to heat stress, not the absolute temperature.
- 5)Line 114-115: Are these annual averages? It would be helpful to report out seasonal averages (wet/dry) so that this work can be easily compared to the other work you referenced in marginal, variable environments. Are the Rock Islands hotter/cooler than the other sites in Samoa, FL, and the GBR?
- 6)Line 176-77: Do you know how much heat stress there was during past ENSO events? If not, you can't just assume thermal stress was comparable between the inner and outer sites.
- 7)Line 182-83: Reword
- 8)Line 191-92: thermal anomaly?
- 9)Line 186-198: How were these data averaged? What years were averaged? Did you include stress bands in your estimate? Growth data that include recent bleaching years may muddle inshore-offshore comparisons; i.e., if using bleaching data, you don't know what growth is like in contrasting environments during benign conditions. The script could flip
- 10)Line 196-98: Where are the bleaching data? There is a lot of discussion of past bleaching, but is there actually bleaching observations that support the inferences from density bands?
- 11)Line 205-206: you don't have actual bleaching data that I can see, thus your interpretation is inferred. Thus, this statement is too strong without monitoring data to support it
- 12)Line 225: Seems to contradict statement in abstract
- 13)Line 251 onward: What about nutrients? Co-author has done a lot of work on nutrients and their influences on growth. What is going on with nutrients in the Rock islands? What about salinity? Its lower salinity, higher nutrients, right?
- 14)Line 306: LB only dominate in channel; this suggests it does NOT thrive at highest heat
- 15)Line 473 – this is the biggest hiccup of the paper. What is the time frame of the analyses? It is very disingenuous and disappointing to present growth this way

Reviewer #3 (Remarks to the Author):

Rivera et. al present a manuscript focussed on coral cryptic diversity and thermal tolerance in Palau. The authors combined a range of genetic and historical data to explore the spatial variation and differences in thermal tolerance among cryptic lineages across Palau. They identify a particular lineage that is more common to the marginal Rock Island (LB) sites and that shows reduced stress levels during previous bleaching events. The results also suggest no habitat effect on growth in this lineage. In general, the manuscript is clear, well-written, and explores a timely topic in conservation biology. I would also like to commend the authors on integrating coral core data into analyses of cryptic genetic diversity and thermal tolerance...very cool and novel. There are, however, points that need major clarification before this manuscript would be suitable for publication.

1. Assigning lineages: It is unclear how the authors deal with admixed individuals. I would suggest removing them from the analyses. What is the threshold to confidently assign each sample a specific genetic lineage? There seems to be a significant amount of admixture, which may be worth discussing more. Please outline this process clearly in the methods.

2. Gene flow: The authors touch upon levels of divergence among the lineages, but make no attempt to explore genetic structure within each lineage. It seems there are enough samples to estimate gene flow between sheltered and exposed sites. This information would be very informative in interpreting the results. They mention the results from particle tracking models, but perhaps exploring this with the genetic data might provide some insight? What is the reproductive mode of *P. lobata*? Would we expect high gene flow between habitats? If so, how would this change our interpretations of results. RI islands are not self-recruiting, but select for tough lineages from a common gene pool?

3. Trade-offs: The authors state no clear trade-offs with gains in thermal tolerance in the LB lineage, although I think this claim may need to be revisited. My interpretation of the data is that there are apparent trade-offs, and that the thermal tolerance of the LB lineage comes at a cost, which is why they are likely outcompeted on the reef slope. Growth rates (particularly skeletal density) are lower in LB than DB corals. The RI sites are effectively selecting for slow growing, but thermally tolerant corals. No trade-offs with gains in thermal tolerance would manifest as no differences in growth between LB and DB lineage.

I believe that once these points are thoroughly addressed, the manuscript will be suitable for publication. Nice paper. Cheers

Luke Thomas

Line numbers in third column referred to clean updated manuscript

Reviewer 1

Reviewer Comment	Response	Line #s
This manuscript presents the findings of Rivera and colleagues who studied bleaching patterns in Porites cf. lobata across large environmental gradients in Palau. The authors found that Porites cf. lobata is comprised of at least 4 'lineages' whose abundances are unequal across geographic sites, with some being more common in Palau's rock islands while others are more common in outer reef environments. These lineages are also biologically differentiated in terms of their calcification rate, extension rate and stress band prevalence (a proxy of bleaching stress), largely driven by the 'RD' lineage. Overall this paper is a solid contribution to the field, in particular the joint analysis of growth rates with bleaching stress and the consideration of adaptation to smaller-scale environmental variation within an island archipelago, which is particularly lacking in marine studies.	We thank the reviewer for their comments, and their positive assessment of our work.	NA
The statistical tests were appropriate, although I would suggest the authors perform additional analyses to identify patterns of local adaptation within each lineage, which would complement their approach describing differences between the lineages.	We thank the reviewer for this suggestion, and we have provided a longer response to this point below.	NA
The authors could strengthen the manuscript by more carefully considering their results across different levels of biological organization. My impression is that the authors have chosen the word 'lineage' to avoid wading into a discussion on cryptic species versus immense levels of population genetic structure, but in the process of trying to remain agnostic the analysis and discussion have lost a great deal of clarity.	The reviewer raises an interesting point. Please see our response below.	NA

Furthermore, a more careful consideration of the role of hosts and symbionts in this system would greatly improve the manuscript, as this species is one where host-symbiont coevolution and local adaptation in either partner could dramatically alter the bleaching response of the holobiont.	Thank you for this suggestion. In the revised manuscript, we have added detail on symbiont genetic patterns in Figure S3 and in the results/discussion. See our response below.	186-191; 290-300; Figure S3;
The authors have chosen to remain agnostic about whether these lineages represent cryptic species because they were unable to test the biological species concept (l. 144-147). This choice is the most conservative, it's true that there have been no tests of successful matings between the lineages, the gold standard for defining species on the criteria of reproductive isolation. However, the genetic patterns that the authors present in this study beg the question of what is maintaining very high levels of F_{ST} (~0.7!) in sympatric populations. Unfortunately, much of the discussion hinges on this designation and its ambiguity leads to a lack of clarity about the ecological and evolutionary consequences of bleaching resistance, local adaptation and growth tradeoffs in this (or these) species. For example, beginning on l. 221 "Limited connectivity between the Rock Island and outers reefs as suggested by hydrodynamic model output likely facilitates local adaptation, which is corroborated by the population differentiation seen here." Local adaptation is a population-level phenomenon, and it is therefore important to identify and characterize what constitutes a population. If the lineages represent subpopulations, then this is a reasonable interpretation, but the authors should consider what is maintaining such high levels of genetic differentiation in the Discussion. If the lineages are distinct species, then local adaptation will occur in subpopulations within each lineage, and statistical tests of this hypothesis should be	Since the submission of this manuscript, our team has conducted the first successful ex situ spawning, fertilization, and larval rearing of Porites lobata. In April 2022, we observed synchronous spawning and successful cross-fertilization among individuals belonging to at least two lineages from RI and OR sites. While this is not a full test of the biological species concept, it strongly suggests that the lineages we observed in this study constitute populations rather than cryptic species that are reproductively isolated. Our interpretation of lineages as populations in the manuscript is therefore supported. In response to this comment, we have added a sentence describing our April 2022 observations and a broader discussion of pre- and post-settlement barriers that could lead to divergence between populations in the results and Discussion.	165-173; 251-262

carried out on individuals from the same lineage living in different habitats.		
The discussion of locally adapted lineages would also be improved by more carefully describing the level (intra- vs. inter- lineage) of variation that is being discussed. On line 208, for example, “the Rock Island-associated LB lineage maintains higher thermal tolerance across habitats, without showing any corresponding trade-offs in growth metrics, indicating this lineage may be particularly well suited for restoration” requires an explanation of why this lineage hasn’t become dominant in all habitats if it grows quickly, doesn’t bleach and experiences no tradeoffs. If the lineages are subpopulations, the question is why these traits don’t sweep to fixation. If the lineages are different species, why does one not competitively exclude the others? In some ways these are similar questions, but their distinction matters from an ecological and evolutionary standpoint, and will have more immediate consequences for conservation and management decisions. If lineages do not interbreed, for example, then intervention strategies that attempt to breed the traits of LB into another lineage will not be possible.	We have added a sentence to the Discussion to specify that it is unclear why favorable traits associated with the LB lineage would not become fixed, but that there could potentially be trade-offs in traits we did not measure. See also responses below regarding intra-lineage variation.	367-370
Additionally, the authors point out that the LB lineage might be good candidate for restoration (l. 310), but this ignores the potential ecological significance of RD, DB and PI, especially if they occupy different environmental niches, as discussed on l. 280-281. Furthermore, if the lineages cannot interbreed, local adaptation would occur independently within each lineage, and the contrasts to which the authors draw the reader’s attention (Figs. 2-3 showing differentiation between the lineages; l.223) are unsuitable. Instead, Fig. 5 would be a better example of patterns consistent with local adaptation to the Rock Island vs. Outer Reef	The RD and PI lineages are less attractive for restoration efforts because they are restricted in distribution to primarily Rock Island or outer reef sites, respectively. The DB lineage has lower thermal tolerance and therefore would not be attractive for restoration. We have specified this in the discussion. As described above, there is synchronous spawning and cross-fertilization of gametes between at least two lineages. These observations strongly suggest that the lineages can interbreed, and our discussion of local adaptation between (rather than within) lineages is most appropriate. Furthermore, the existence of admixed individuals within our dataset supports the idea that lineages can	NA

habitats within each lineage (but not sufficient by themselves without experimental follow-up) if each lineage is independent and has subpopulations in these two habitats.	interbreed.	
One analysis that would add evidence for within-lineage local adaptation is to identify population structure in the lineages themselves between the Rock Island and outer reef sites. This would reveal patterns consistent with within-lineage local adaptation between populations inhabiting different environments, for instance population structure between Rock Island and outer reef sites in LB and DB. Growth and bleaching are likely impacted by many underlying genetic mechanisms as well as environmental mechanisms. The authors point this out in A. hyacinthus in American Samoa, but should consider contextualizing their results in light of recent work highlighting a similar pattern of bleaching resistance associated with increased reef temperatures in A. hyacinthus in Palau (Cornwell et al. 2021).	See response below regarding genetic variation within lineages. The back reef pools of American Samoa are the closest analogue to our Rock Island sites, so it is most appropriate to compare our findings to these previous studies. The Cornwell 2021 paper focused on patch reefs in Palau and found quite limited support for trade-offs (conclusions were based correlating two correlations all with very low R²) so we are hesitant to use rely on their finding. There is a robust body of work around the differences in thermal tolerance in Samoa and felt more comfortable using this as our reference point.	NA
If 'lineages' are cryptic species, it could be that each lineage has to contend with a different set of mechanisms (invisible to us until we figure out what to measure) that might lead to trade-offs, but that might only be evident when examining each lineage by itself. Emerging lines of evidence for tradeoffs are now in the literature, including a growth/symbiont density trade-off (Cornwell et al. 2021) that also might impact bleaching severity (Cunning and Baker 2013).	As stated above, we observed synchronous spawning and cross-fertilization in at least two lineages from RI and OR sites, strongly suggesting they are not cryptic species. We therefore focus the discussion on patterns between (rather than within) lineages. While the reviewer is correct that some recent studies have shown trade-offs with thermal tolerance, it is worth noting that the two papers they name both focus on branching species, while P. cf. lobata is mounding, so selective pressures and trade-off dynamics may be slightly different. We further assess that the analysis in the Cornwell paper is based only on weak correlations and offers no mechanistic explanation or support for its trade-off claims, so we choose not to cite the Cornwell paper. We have treated the question of trade-offs carefully in the Discussion. The compound	346-370

	stressors of low pH and aragonite saturation in addition to higher temperatures in Rock Island sites provide a plausible explanation for the lower growth and skeletal density we observed in DB and RD Rock Island corals. Our study provides no solid evidence of a growth trade-off, but future studies could address this experimentally, which we have stated in our discussion.	
On the other hand, the discussion should have a completely different focus if these lineages represent individuals that are all from the same species, instead of cryptic species. What exactly is keeping them so genetically differentiated? Given that the SNPs are from a RAD dataset, one would expect most of them to be selectively neutral, the same goes for the microsatellites. It is difficult to come up with an explanation for what would maintain 'lineages' that are identified using neutral loci without invoking some sort of between-lineage reproductive failure (either pre- or post-zygotic). The authors advance the possibility that there are hybrids (l. 143), an interesting hypothesis to be sure, and one that the authors might be able to bolster using other analytical techniques (e.g. F4 statistics; Reich et al. 2009 Treemix; Pickrell and Pritchard 2012).	The Rock Island sites included in this study are semi-enclosed lagoons. Previous studies using hydrodynamic models have suggested limited connectivity between Rock Island and outer reef sites. It is not necessary to invoke reproductive failure to explain the population differentiation we observed. Limited oceanographic connectivity between Rock Island and outer reef sites could limit larval dispersal, with most larvae settling near their natal reef. Warmer and more variable temperatures at Rock Island sites could also lead to differential mortality in larvae and recruits. Selection for thermal tolerance in the early life-history stages could drive the population differentiation we observed. These concepts are now explained in the discussion. We had in previous iterations of analyses conducted more in-depth demographic analyses using dadi to infer demographic patterns and lineage divergences. We found that the uneven sample sizes between lineages substantially limited the interpretability of those results as thus have not presented treemix, dadi, or other similar demographic analyses here. As we are continuing work with these populations we feel it is better to obtain more appropriate sample sizes and spatial distribution of samples to adequately address this.	NA
It's worth mentioning that the lineage with very high levels of FST (PI) only looks admixed in one individual, and that individual has ancestry from all 4 lineages (Fig. 3 11th bar from the left in the Risong population) which suggests ambiguity in its assignment as a	There are several other admixed individuals with some contribution from the PI lineage at Risong (4th bar from the left), Helen (3rd bar from the left) and Ngerchelong (last bar on the right). Nevertheless, it is true that admixture is uneven among lineages. There is also more evident admixture in the microsatellite STRUCTURE results (Figure S1). We have	163-167

possible alternate hypothesis to an admixed origin.

acknowledged this more clearly in the Results. The higher levels of Fst between this lineage and others may also be driven by the smaller sample size and a majority of the PI samples originating from the most distant sites (Ngerchelong, Kayangel, and Helen). As such the site/distance driven population structure would contribute more so to the between lineage differentiation we're assessing. We've now discussed this more clearly in the results.

Finally, the contribution of the symbionts to the patterns described in the study deserves more discussion, especially given that this coral species transmits them vertically from mother to offspring. Vertical transmission is a very effective way of aligning the fitness interests of both the hosts and symbionts, which means that local adaptation in the symbionts can be highly beneficial to the host in the current generation as well as future generations where host-symbiont lineage combinations will remain intact. Treating the symbionts as a homogeneous population devoid of variation is inappropriate on the basis of a single genetic locus. At best it ignores neutral variation, at worst the symbionts are major drivers of these patterns, but that variation remains hidden. It is challenging to generate multilocus genomic datasets for the symbionts, but recent studies suggest that they are highly genetically structured by geography (Thornhill et al. 2017; Cooke et al. 2020; Kriefall et al. 2022), host lineage (Forsman et al. 2020); possibly by environment (Klueter et al. 2015; Cornwell and Hernandez 2021; Kriefall et al. 2022). Many (but not all) of these patterns would be largely undetectable at the ITS locus, and the discussion should consider this when interpreting phenotypic variation and patterns of local adaptation in the coral 'lineages.'

We thank the reviewer for these comments. You are right that the symbionts are an interesting component to look at here. When we originally started this work and were just using microsatellites, we relied on DGGE to get some preliminary insights into the symbiont composition. While we agree that there could be within C15 variation between our samples, our first interest was to ensure that there weren't different species of symbionts between environments.

With the RAD data, we are able to look at the symbiont diversity a bit more closely. We ended up with only a small number of RAD SNPs that passed all our filtering criteria (218 SNPs), so it's a bit limited in that regard. Just looking at those SNPs, though, we do very nicely recapture clusters that match the host coral lineage. New figure:

186-191;
290-300;
Fig S3

	This is both expected and a bit confounding. As you rightly mentioned, symbionts in this species are vertically transmitted. Given the strong population differentiation between lineages that the coral SNP data show, it would be reasonable to also see strong differentiation in the symbionts. So we could go all the way to one extreme and attribute all the differences in phenotypes between the lineages to the symbionts, or on the other extreme only to the coral host. We see much lower F_{st}s between lineages for the symbiont RAD data (0.04-0.07) compared to the much higher differences between lineages in the coral host SNP data. We do see fairly high F_{st} in the symbiont data for the PI lineage (0.2) relative to other lineages, but this is also consistent with that F_{st} being way higher for the host (0.7). Because we don't seem to have differences in symbiont species composition between our samples, we had leaned fairly heavily towards attributing the phenotypic differences to the host lineage differences, since most of the literature on thermal tolerance differences in symbionts focuses on differences between genera of symbionts (former clades). Nevertheless, we realize that it is not possible simply from these patterns to entirely rule out the potential influences of symbionts (or a possible co-evolution of symbionts with certain host lineages) on growth and thermal tolerance. We have now included more detailed symbiont results in the manuscript (as Figure S3) and discussed these patterns further in our results and discussion.	
The authors should consider explicitly stating that symbionts in this species are vertically transmitted in the introduction.	This has been added to the end of the sentence describing symbiont composition in Porites. "While harboring more thermally tolerant genera of Symbiodiniaceae, such as Durusdinium, can make corals less susceptible to bleaching, massive and branching Porites species throughout the Indo-Pacific nearly exclusively harbor Cladocopium (formerly	108-109

	C15), and these are transmitted vertically from mother to offspring”.	
I. 183 change ‘confined two’ to ‘confined to two’	Corrected, thank you.	181
I. 186 – Why only DB/LB and not PI? Seems like PI has sample sizes in the Rock Islands approaching the sample size of DB in the same habitat.	There is only one PI sample from the RI for which we sampled a coral core. The other PI samples were from smaller coral colonies that were not cored. Hence, we do not have an adequate sample size to compare growth or stress band patterns of PI corals across habitats.	NA
I. 186-198 – The authors should consider reframing this paragraph into two sets of comparisons. First, within-lineage comparisons (e.g. LB in rock islands compared to LB on outer reef). Second, between-lineage comparisons. Are there differences between LB and DB in both places? These two sets of comparisons would set up an analysis for examining local adaptation (see Kawecki and Ebert 2004 for a well-reasoned discussion of how to approach these analyses). This is important, it looks like LB performs the same or worse than DB on outer reefs which points to potential tradeoffs that the authors argue against later in the text (I. 293). If there are no tradeoffs the statistical comparisons that inform that conclusion should be clearer, and I would suggest a power analysis to determine what effect size could have been detected with these sample sizes.	We have followed the reviewer’s suggestion by leading the paragraph with within-lineage comparisons and then highlighting the different patterns between LB and DB corals exposed to different environments. We have also conducted a power analysis to evaluate the probability of Type II error in our results. We calculated the effect size for each growth parameter (density, calcification rate, extension rate) using Cohen’s F and then used these values to estimate the number of replicates needed in each site-lineage group to achieve a power of 0.8. The power analyses showed 6, 11, and 16 replicates needed for density, calcification, and extension rate, respectively. Our lowest sample size was N=6 (for DB corals in the Rock Islands), so we had sufficient replication for skeletal density but not the other parameters. Nevertheless, the power analysis is actually unnecessary because ANOVAs revealed significant differences between groups for each of the three growth parameters. Type II error (false negative) is therefore not as much a concern, since we did find significant differences. In the last paragraph of the Results section, we have reported the results of our ANOVA and post hoc Tukey tests, which showed significant differences in skeletal density and calcification rate for DB corals from the Rock Islands versus the outer reefs. No significant differences in growth were detected for LB corals. These results support our conclusion that there are no apparent trade-offs to thermal	219-232; 346-370

	tolerance in LB corals, and that growth impacts are inconsistent across lineages. It is still important to keep in mind that the most parsimonious explanation for growth impacts in Rock Island corals is not a trade-off with thermal tolerance but the compound stressor of low pH. Previous research has consistently shown that water in the Rock Island lagoons is more acidic than the surrounding open ocean and that low aragonite saturation leads to lower calcification in corals. We have addressed this clearly in the Discussion, while acknowledging that other studies should address the question of trade-offs directly.	
I. 204 – Rephrase ‘promoted the development of thermally tolerant lineages.’ Do you mean that the Rock Islands create an environment that selects for thermally tolerant lineages? Aside from a lack of clarity it also makes it sound like the Rock Islands are the reason these lineages are physiologically and genetically different, but there is no evidence for that in this study.	We used “promoted” as a way to encapsulate both selective pressures and/or plasticity that could be facilitated by the Rock Islands environments. The Rock Islands have higher temperature diurnal ranges, which has been previously shown to enhance thermal tolerance. This enhancement can happen through the induction of plasticity with or without selection. This sentence has been modified to clarify and now reads: “The unique environmental conditions of Palau’s Rock Islands where higher temperatures can serve as selective pressure and larger temperature ranges can facilitate plasticity have likely promoted the development of thermally tolerant lineages (Fig. 1).”	
I. 208 – Specify if ‘driven by genetic factors’ means within the host, symbiont or both. See above for why it could just as easily be the symbionts given that they are vertically transmitted.	This now reads "driven by genetic factors in the host and possibly the symbiont." We have added more discussion of the population structure in the symbiont and included analysis of the RAD data. See response above.	243
I. 217-219 – Is it possible that you have more evidence for this in the core data? Do those samples allow you to see that colonies settled after the last heat wave? It could be an interesting addition if individuals from less stress tolerant lineages are all young and settled after the last heating event, the hypothesis being that they have yet to experience a strong selective filter.	This is an interesting point. Our 2018 samples included small coral colonies. We measured two diameters (long and short axes) for each colony and used these data to estimate ages. Guzman and Cortes (1989) reported a maximum of 2 cm annual radial growth for Porites lobata, so based on this, most of our 2018 specimens would be expected to be 1.5 - 4 years old. They would thus be expected to	

	have recruited after the 2010 ENSO, and some individuals may have recruited after the 2015 ENSO. One of the preliminary analyses we conducted in writing this manuscript was comparing lineage assignments for large v. small colonies, and the results are qualitatively similar for all sites. This suggests that selection acts on the early life-history stages - before the corals are a few years old, likely on larvae and newly-settled recruits. We have not reported our age estimates for small coral samples in the manuscript because the analysis is based on growth rate estimates from a very different environment (Costa Rica) and therefore potentially inaccurate for Palau. However, we are confident in describing the smaller coral colonies we sampled as “likely only a few years old” and have added a discussion of the potential for selection to act on the early life-history stages to the manuscript. Citation Guzman and Cortes (1989) Growth Rates of Eight Species of Scleractinian Corals in the Eastern Pacific (Costa Rica) Bulletin of Marine Science, Volume 44, Number 3, May 1989, pp. 1186-1194(9)	
I. 221-223 – Can’t your genetic data shed some light on whether there are high levels of gene flow between the rock islands and the outer reefs? If the Rock Islands are selecting for only thermally tolerant individuals you would expect higher rates of gene flow between rock island sites than between rock island and outer reef sites within each lineage.	We thank the reviewer for this suggestion. We have now looked at the levels of gene flow (using F_{st} and N_m as estimated with F_{st}) within each lineage across Sites. For some lineages the information is not very valuable as there are very few individuals or none from that lineage from several sites. For the LB lineage, we see the expected pattern where the number of expected migrants is higher between RI sites than between RI and OR sites (and also between OR sites). These results are consistent between RAD and microsat derived values. These patterns are corroborated in PCAs with RI samples clustering more closely together than OR samples.	NA

For the DB lineage, the patterns are a little different. With RAD data there are fewer migrants between RI sites than other pairings. However, there are several sites with only 1 DB individual, and several sites with fewer than 4 individuals, so these estimates are really unideal. For microsat data, we have much better sample sizes. We still see higher F_{st} and lower migrants between RI sites than between OR sites, but there is still lower migration between OR and RI sites than between RI sites.

Given that DB corals are more common on outer reefs and there are fewer barriers to dispersal across the open outer reefs, it is also reasonable that between outer reefs F_{st} s would be lower. The lower F_{st} /higher migration between RI sites relative to between OR and RI sites could indicate similar selection pressures across RI sites, which is interesting.

With the PI and RD lineages, neither the RAD or microsatellite data is very useful, as many sites are not represented in either dataset. The PCAs are also fairly uninformative. See below.

Please note that the PCA graphs for RAD and microsatellite data are not showing all the same samples. Some samples were processed with RAD, others with microsatellites, and some with both (N=73). There is some overlap in samples, but there are unique samples in each set.

	 LB lineage, RAD data LB lineage, msat data DB lineage, RAD data DB lineage, msat data PI lineage, RAD data PI lineage, msat data RD lineage, RAD data RD lineage, msat data Region ■ Outer Reefs ▲ Rock Islands While we appreciate the reviewer’s suggestion, we think that the uneven representation of lineages across sites makes relying on any of these Fst or Nm estimates a bit questionable. So we prefer to keep the presentation in our manuscript as is and focus on the thermal tolerance and growth differences between lineages instead of delving into specific demographic estimates.	
I. 230 – ‘produce’ should be replaced, do you mean ‘select for’ or are you advocating that the Rock Islands are the actual location where these thermally tolerant lineages originated? I’d believe the former but not necessarily the latter (absent much more data from across the geographic range of the species).	We have replaced the word “produce” with “select for.”	283

I. 236 – See comment for I. 214-215, the comparison to American Samoa A. hyacinthus populations is apt but comparing to Palau populations seems even more relevant.	We don't see any comments for line 214-215. Nevertheless, we chose to keep the citations currently in the manuscript because the studies cited compare thermal tolerance of A. hyacinthus corals from a warmer and thermally variable environment, which is a direct analogue to our study. Previous research on A. hyacinthus in Palau by Cros et al. did not tie genetic patterns to temperatures at sampling sites directly, and we have some hesitations regarding Cornwell et al. 2021, the only other A. hyacinthus paper from Palau that we are aware of. Citation Cros, A., Toonen, R. J., Donahue, M. J., & Karl, S. A. (2017). Connecting Palau's marine protected areas: A population genetic approach to conservation. Coral Reefs, 36, 735–748. https://doi.org/10.1007/s00338-017-1565-x	NA
I. 274 – Is it fair to consider diverged lineages that occur at different depths 'sympatric'?	Sympatry refers to a situation in which there are no physical barriers preventing members of a species from mating with one another. Corals occurring at different depths at the same site have no physical barriers to reproduction, although admittedly there are potentially oceanographic factors that could reduce the probability of cross-fertilization. The answer also depends on reproductive mode of the species in question, since hermaphrodites tend to have highly buoyant packets of gametes while gonochoric species have more neutrally buoyant eggs and sperm that are distributed throughout the water column. Porites cf. lobata is a gonochoric broadcast spawner. In April 2022, we conducted the first ex situ spawning and larval culturing of this species. We observed both eggs and sperm distributed throughout the culture dishes, indicating relatively neutral buoyancy. There is a possibility that gametes from deeper individuals would not encounter gametes from shallower individuals because of this, but the depths of our sampling sites (< 6 m) presents a rather constrained water column that would	NA

	increase the chance of cross-fertilization. The sentence the reviewer is referring to in this comment is a very small note in the discussion, so we choose not to make substantial edits to the manuscript to address the broader question they have posed. In principle, two individuals that occur at different depths at the same site with no physical barriers preventing reproduction between them could be described as sympatric.	
I. 305-307 – Consider adding a discussion of what the appropriate spatial scale is to be asking about local adaptation in these populations. Outer reef and Rock Islands habitats are clearly different, but are there microhabitats in each that provide refuge for thermally (in)tolerant lineages? One reason why LB looks unaffected across habitats could be that it is good at finding ideal microhabitats in each location.	This is a very interesting thought and we have regretted not carefully mapping the location of all the corals sampled in order to investigate if there is some microhabitat structuring that explains these co-existing lineages as some kind of environmental niche partitioning. We are continuing some of this work and keeping these potential questions in mind for future collections. While it would be really fascinating to find these microhabitat patterns, on the other hand it may also be very difficult to actually be certain that there are differences between microhabitats that are driving niche specialization. Conditions on reefs can vary temporally as much or more than spatially and certainly the actual structure of the reefs can be heavily impacted with storms or other physical damage. We have included some discussion of other mitigating factors that may facilitate thermal tolerance in the RI in particular, such as shading or increased heterotrophic opportunity. We've now also added a sentence on the possibility of other microhabitat features as food for thought in our discussion, but we think that going any further in implicating microhabitats as a reason for these differences would be purely speculative.	307-311
I. 321 – Rephrase 'increase selective pressures.' Do you mean that long residence times ensure that larvae from each generation tend to land in the same geographic location as their parents, thus exposing them to	Yes, the reviewer has interpreted our meaning correctly. However, the paragraph in which this sentence appeared has now been removed in response to another comment. Therefore, rephrasing this sentence is no longer required.	NA

consistent selection pressures across generations?		
I. 321-323 – Without more population genetic data this really seems like a stretch. If you are going to make the argument that they originated in the rock islands you should show more data to bolster it. There is no evidence presented here that the Rock Island corals are a closed population, and have been that way for 30-50 generations.	The reviewer raises a very valid point. We have removed this paragraph from the discussion.	NA
I. 403 – RAD data are from single end reads, does this mean that the authors could not detect and remove PCR duplicates arising during library construction? It leads me to wonder if this is partly driving the exceedingly high levels of FST between the lineages. The agreement between the RAD data set and microsatellites implies that it isn't playing such a large role in identifying members of each 'lineage' but perhaps is inflating estimates of differentiation if heterozygotes are missed. Or was there some sort of random identifier in the barcodes that allowed the authors to overcome this problem?	The method used here does not allow for the identification or removal of PCR duplicates. While the very high Fst seen in the RAD may be artificially inflated by the presence of PCR duplicates, as the reviewer points out, the extremely high agreement between microsatellite and RAD results in population structure suggest that any influence of PCR duplicates is not influencing our patterns. In addition our analyses do not rely on using the actual values of Fst for inferring other population genetic metrics such as effective population size, etc. Therefore we do not think that the inability to remove PCR duplicates impacts any of our conclusions. To provide a bit more context on the RAD methodology: there are various methods for conducting RADseq (e.g. single enzyme RAD, ddRAD, 2bRAD, mbRAD, etc.). The only one that allows for the detection of PCR duplicates is mbRAD, which is much more technically complex and expensive to conduct than other RAD options. mbRAD is much less common in large genetic datasets. In addition, all previous coral studies, to our knowledge, that have used RAD use one of the methods for which PCR duplicates cannot be removed. Our methods and analyses are comparable to other studies and methods in coral genetics.	NA
I. 431 – remove 'using'	Change made.	505
Fig. 1B – Change 'Nikko' to 'Ngermid' to stay consistent with I. 343	Thank you for catching that! Fixed.	Figure 1

Fig. 5 – Can you use the same visual language of ‘*’ to show the statistical comparisons you made and whether they were significant as in Fig. 4? This is especially important for making an argument about local adaptation within each lineage. Also, to reiterate a comment from above, why not include PI in these plots?	Figure 5 has been updated to include indications of statistical significance between site and lineage groups. There is only one PI sample from the RI for which we sampled a coral core. The other samples were from smaller coral colonies that were not cored. Hence, we do not have sufficient sample size to compare growth or stress band patterns of PI corals across habitats.	Figure 5
l. 611 – Consider changing ‘colored by their dominant’ to ‘colored by their dominant lineage’	Change made.	Figure 3
l. 764 and 766 – Are there peer reviewed versions of these publications?	No, unfortunately, there do not appear to be peer-reviewed versions of these publications. However, each publication concerns a method for analysis of bioinformatic data, so they could be considered equivalent to R packages that are also citable but not peer-reviewed.	NA

Reviewer #2

Reviewer Comment	Response	Line #s
This is an important study that falls short in the presentation of the data. It feels as though the authors are trying to pull more from their paper than what the data supports. This is unfortunate because the paper stands on its own, but the authors need to reign in (or better present) some of their claims, as well as provide more information on growth analyses.	We thank the reviewer for their comments and aim to publish a solid, well-grounded paper.	NA
First, the authors need to include discussion about the drawbacks of using massive Porites as their organism for study. This group of species are naturally very stress tolerant. In fact, they tend to be one of the few species that is always found in very marginal environments. Massive Porites aren’t building reefs like acroporids, except in the most extreme of environmental conditions that limit the presence	What the reviewer has portrayed as “drawbacks,” we see as strengths. It is true that mounding corals such as Porites lobata are generally less susceptible to stressors and present in marginal habitats. This is exactly our reason for selecting it as our study species. The resilience of Porites corals provides clues to understanding the future of coral reefs, particularly mechanisms of adaptation, plasticity, physiology, and ecology	85-94

of all other coral species. Furthermore, this species possesses a very porous skeleton that allows the coral to retract its tissues deep into the skeleton during times of stress, including bleaching. The discussion needs to be written so that the authors acknowledge and defend using a species of coral that will likely see the smallest population decline of any coral on the planet with global warming.

that can help maintain coral reefs in the face of environmental stress. Mounding corals also have the distinct advantage that bleaching records can be garnered from individual colonies by coring, so we are able to tie genotype to bleaching history for multiple individuals across our study sites. We have now explained our reasoning for selecting *Porites lobata* as our study species in the introduction.

In addition, massive corals like *Porites* are absolutely reef building and can even grow large enough to build micro-atolls. The extremely large skeleton is shaped and eroded over its centuries of life to create habitats for all varieties of fish, and invertebrates seek refuge in its dome from larger predators. Massive species are essential for reef fish and invertebrates and serve a critical function in coral reef landscapes (Nanami and Nishihira 2004; Sloan 1982; Peyrot-Clausade et al. 1992; Wilson et al. 2013).

Lastly, for the reasons the reviewer states, *Porites* corals will likely dominate future reef environments. We see this as an incredibly important reason to understand their thermal tolerance and population connectivity.

More context on these points has been added to the introduction.

Citations

Nanami, A., Nishihira, M. Microhabitat association and temporal stability in reef fish assemblages on massive *Porites* microatolls. *Ichthyol Res* 51, 165–171 (2004).

Sloan, N.A. Size and structure of echinoderm populations associated with different coexisting coral species at Aldabra Atoll, Seychelles. *Mar. Biol.* 66, 67–75 (1982).

Peyrot-Clausade, M., Hutchings, P. & Richard, G. Temporal variations of macroborers in massive *Porites lobata* on Moorea, French Polynesia. *Coral Reefs* 11, 161–166 (1992)

	Wilson, S.K., Fisher, R. & Pratchett, M.S. Differential use of shelter holes by sympatric species of blennies (Blennidae). Mar Biol 160, 2405–2411 (2013)	
Second, the discussion of temperatures and thermal stress between the rock islands and offshore sites is not clear. Phrases like thermal anomaly and heat stress are used without any quantification of what the thermal anomaly or thermal stress actually is. The absolute temperatures, while interesting and show us that the corals persist in a warmer environment, tell us nothing about thermal stress as it pertains to the anomaly from climatology, or average conditions. Just because it's hotter in the Rock Islands does not mean that these sites experience more thermal stress, nor does it mean that these sites experienced the same levels of thermal stress and bleaching that was reported on the outer reefs in the past. What are the climatologies and thermal anomalies for the offshore versus inshore reefs? What about the history of thermal stress for these two locations? This is not discussed, nor expounded on. It is the thermal anomaly, or deviation from the climatology or long-term averages, that matters when it comes to bleaching.	There have been multiple studies looking at the thermal anomalies and temperature patterns across the Palauan archipelago during the 1998 and 2010 ENSO events (listed below) and which are cited in our introduction where we describe these differences. For instance, in 2010 several of the Rock Island sites in this paper experienced anomalies of up to 2C, while outer reefs experienced either negative anomalies or up to only 1 C, relative to a 2001-2011 average. (Van Woesik et. al 2012). Bleaching patterns and community composition across Palauan reefs have also been well documented since 1998, and inner bay reefs have consistently shown lower bleaching, faster recovery, and minimal shifts in overall community composition, further suggesting that they have higher resistance/resilience to bleaching events (Golbuu et al. 1998, Barkley and Cohen 2016, Van Woesik et al. 2012, Bruno, et al. 2001, Gouezo et al. 2019) Since the focus of this paper is on the genetic patterns of corals across these habitats, we did not repeat the longer temp climatological analyses that have already been conducted by others in these and similar sites. In addition, part of our shown temperature data was graciously shared by Pat Collin who has operated multiple temperature loggers in Palau for quite some time. Upon requesting the data we mentioned that we would be using it for exploratory purposes and not conducting rigorous analyses. (He has seen our manuscript and agreed to our use of his data). As such, our Figure 1 data simply serves to illustrate for the reader the temperature patterns across most of the sites	133; 140-142

	sampled in this paper. We intend for it to serve as supporting information for our discussion of temperature differences in combination with the several cited studies on these patterns. We have added language clarifying this use on Lines 140-142 We have also changed the word “chronically” to “consistently” in Line 133 to better suggest that this 2C difference is based simply on the data presented in Figure 1 as opposed to being supported by longer term climatological averages. Citations Bruno, J. F., Siddon, C. E., Witman, J. D., Colin, P. L. & Toscano, M. A. El Niño related coral bleaching in Palau, western Caroline Islands. Coral Reefs 20, 127–136 (2001). Golbuu, Y. et al. Palau’s coral reefs show differential habitat recovery following the 1998-bleaching event. Coral Reefs 26, 319–332 (2007). Gouezo, M., et al. Drivers of recovery and reassembly of coral reef communities. Proceedings of the Royal Society B: Biological Sciences, 286 (2019). van Woelik, R. et al. Climate-change refugia in the sheltered bays of Palau: analogs of future reefs. Ecol. Evol. 2, 2474–2484 (2012). Barkley, H. C. & Cohen, A. L. Skeletal records of community-level bleaching in Porites corals from Palau. Coral Reefs 35, 1407–1417 (2016).	
Third, density bands are not fully accepted by the sclerochronology community as a means to ID past bleaching events. I realize there was a published study – but this was from an ENSO dominated location with huge thermal anomalies that is an outlier to what most reefs experience. You can’t use density bands as the	We agree that annual skeletal high-low density band couplets, the basis for coral based sclerochronology, do not provide proof of bleaching. However, we specifically refer here to anomalously high-density stress bands, accreted during heatwave events (see below image). These stress bands are anomalous both in the timing of their formation	98-103

only evidence for past thermal stress and bleaching.

within the annual high-low density band cycle, and in the exceptionally high density of the band relative to “normal” non-stress high density bands.

Contrary to the reviewer’s assertion, such high-density stress bands in *Porites* and other massive corals have been widely used to document past bleaching events (see Lough and Cooper 2011; Cantin et al., 2010; Carilli et al., 2010; Carille et al., 2014; Cantin & Lough, 2014; Barkley & Cohen, 2016; DeCarlo et al., 2017; DeCarlo et al., 2019; DeCarlo et al., 2020; Barkley et al., 2018, Barkley & Cohen, 2016; Mollica et al., 2019; DeCarlo et al., 2020).

Further, recent studies uncovering the mechanism of stress band formation established the link between bleaching-induced starvation and the consequent reduction in skeletal extension. The authors refer the reviewer to two recent papers describing the mechanism of stress bands formation in both ENSO dominated and non-ENSO dominated locations: DeCarlo and Cohen 2016, and Barkley et al., 2017.

The additional citations listed here have now been incorporated throughout the manuscript as appropriate to provide additional support of our methods for the reader’s interest.

Citations

Cantin, N. E. & Lough, J. M. Surviving Coral Bleaching Events: *Porites* Growth Anomalies on the Great Barrier Reef. PLOS ONE 9,

e88720 (2014).

Carilli, J. E., Norris, R. D., Black, B., Walsh, S. M. & Mcfield, M. Century-scale records of coral growth rates indicate that local stressors reduce coral thermal tolerance threshold. *Global Change Biology* 16, 1247–1257 (2010).

Cantin, N. E., Cohen, A. L., Karnauskas, K. B., Tarrant, A. M. & McCorkle, D. C. Ocean warming slows coral growth in the central Red Sea. *Science* (1979) 329, 322–325 (2010).

Lough, J. M. & Cooper, T. F. New insights from coral growth band studies in an era of rapid environmental change. *Earth-Science Reviews* 108, 170–184 (2011).

Mollica, N. R. N. et al. Skeletal records of bleaching reveal different thermal thresholds of Pacific coral reef assemblages. *Coral Reefs* 38, 743–757 (2019).

Barkley, H. C. et al. Repeat bleaching of a central Pacific coral reef over the past six decades (1960-2016). *Communications Biology* 1, Article Number: 177 (2018).

Barkley, H. C. & Cohen, A. L. Skeletal records of community-level bleaching in *Porites* corals from Palau. *Coral Reefs* 35, 1407–1417 (2016).

DeCarlo, T. M. & Cohen, A. L. Dissepiments, density bands and signatures of thermal stress in *Porites* skeletons. *Coral Reefs* 36, 749–761 (2017).

DeCarlo, T. M. et al. Acclimatization of massive reef-building corals to consecutive heatwaves. *Proceedings of the Royal Society B* 286, (2019).

DeCarlo, T. M. The past century of coral bleaching in the Saudi Arabian central Red Sea. *PeerJ* 8, e10200 (2020).

Finally, the growth data are presented in a disingenuous manner and the data have been massaged to serve the narrative that there aren't growth trade-offs. The authors are reaching for a result that their data do not support. By looking at Figure 4, I'd say there most likely ARE growth trade-offs to living in very hot environments. Lineage RD, which only occurs in the hottest and most thermally variable (i.e., most extreme) environments of Risong and Mecherchar, clearly has very low density, calcification and extension. For the lineage that co-exists across offshore and Rock Islands (e.g., LB), this is only found in the middle of the channel, where there is much more water exchange and not where the environment is most extreme. In other words, they exist somewhere in between the stable, cool offshore conditions and the most hot and variable environments, which restricted lineages.

The reviewer is correct that the RD lineage has lower growth than LB and DB lineages. The reviewer is also correct that the LB lineage is most common at the site "Mecherchar Channel" and uncommon at the more isolated site "Mecherchar." However, the LB lineage also occurs at the Rock Island sites Risong, Taoch, and Ngermid, and it is even the dominant lineage at Taoch and Ngermid. An examination of Figure 4 in isolation may lead a reasonable person to conclude that there are trade-offs between thermal tolerance and growth. However, it is absolutely essential to consider these results in the context of environmental conditions at each site. As we have written in the discussion, Rock Island sites have lower pH and aragonite saturation, which are known to impact skeletal density and increase bioerosion. We therefore cannot conclude that there is a trade-off between thermal tolerance and growth in *Porites cf. lobata*. The principle of parsimony points toward the obvious explanation that growth and skeletal density in Rock Island corals was influenced by the pH and aragonite saturation at those sites.

Furthermore, data for skeletal density, calcification, and extension rate for LB corals from Rock Island sites shown in Figure 5 do NOT include individuals from the Mecherchar Channel (none of these samples were ever cored). We recognized that these individuals were exposed to different environmental conditions than their counterparts at Mecherchar and did not include them in the analysis - specifically to avoid compounding our results as the reviewer suggests we did.

Our data very clearly show no significant differences in skeletal density, calcification, or extension rate among LB corals from Rock Island and outer reef sites. As stated in the discussion, there does not appear to be any evidence of a trade-off between thermal tolerance and growth in this lineage. On the other hand, our data show lower skeletal density, calcification, and extension rate for DB corals from Rock Island sites compared to outer reef sites. This result must still be

	considered in the context of lower pH and aragonite saturation at Rock Island compared to outer reef sites. In the manuscript, we have stated “Though one could interpret the DB’s lower growth as a trade-off with thermal tolerance, the challenging conditions for calcification in the Rock Islands are more likely to be driving factors” and “The RD lineage, which is nearly exclusively found in the two lowest pH sites, Risong and Mercherchar^{14,15}, shows lower growth metrics than the LB lineage, suggesting Rock Island conditions do have the potential to hinder coral growth.” We have not stated definitively that there are no trade-offs between thermal tolerance and growth in corals, nor have we made any attempt to hide or “massage” results to fit a narrative. Rather, we have presented the most logical, parsimonious explanation for the patterns we observed.	
The issue with the core data is that there is zero discussion about how the data were averaged. The authors say they use prior and new core data, but what years do these cover? Did all your cores cover the exact same number of years? If not, the data may be biased. Did you include both known bleaching and non-bleaching years in your averaging of growth rates? What about the number of data points per year per site? I’m guessing the cores at these inshore sites had a lot of alteration, including bioerosion, and that getting data from them may have been more challenging than the authors indicate. The second author on this paper is a world leader in coral core analysis and the way these data are handled and reported is very disappointing. This is a shame because the paper does not hinge on these data at all and stands on its own without needing to cryptically package the growth data to try and claim there is no growth tradeoff	We thank the reviewer for pointing out this lack of detail in our methods. We have included a lot more clarity on the cores in the methods now. The years covered by the cores collected ranged from 1970 to 2014, with most cores ranging from 1994 to 2013. As the reviewer points out, it would be unwise to use growth data from different periods for different cores. The analyses presented in the paper used only the 1995-2014 for growth data (calcification, extension rates, and density). As the reviewer notes also, including bleaching years may also lead to issues. Stress bands were assessed using all the data available for each individual core. This was because the program used for stress band identification analyzes a continuous density transect across the core and identifies areas of growth with anomalously high density (relative to the whole core average density) and an anomalous gradient in density (the change in density from pre/post stress band area is anomalously high). In response to this comment, we have	531-559;

Line 186-198: How were these data averaged? What years were averaged? Did you include stress bands in your estimate? Growth data that include recent bleaching years may muddle inshore-offshore comparisons; i.e., if using bleaching data, you don't know what growth is like in contrasting environments during benign conditions. The script could flip

Line 473 – this is the biggest hiccup of the paper. What is the time frame of the analyses? It is very disingenuous and disappointing to present growth this way

reanalyzed the core growth data to only include the years 1994-2013, removing 1998 and 2010 to avoid bleaching years. The results remain the same as previously.

Original figure 4:

Revised using only 1994-2013 (-1998,-2010) data:

We have added more detail on both the identification of stress bands and the years of data used for growth averages in the methods section on lines 556-559.

The results that were presented in Figure 5 now do not show significant differences for extension rates in the DB lineage in outer reef vs Rock Island habitats. The finding that is more relevant for our paper and our conclusion about the LB lineage remains entirely unchanged: it does not show any differences in growth among habitats.

Original Figure 5:

Updated Figure 5 using only 1994-2013 (-1998,-2010) data:

We have tried to present our growth data in a simple and straightforward way to best portray differences between lineages and across habitats. To better visualize comparison between Rock Island and outer reef habitats in the DB and LB lineages, we have reorganized Figure 5 to have panels by

lineages instead by region. This should enable easier comparisons of the declines in certain parameters for the DB lineage, and consistent growth patterns in the LB lineage.

New version of Figure 5:

Lastly, there are indeed many cores (N=42) from our study that had bioerosion, cracks, or were otherwise difficult to properly age due to inconsistent banding patterns or other issues such as partial mortality scars. None of these cores are included in this analysis. We only include cores for which we could confidently identify growth bands and obtain well supported estimates of growth, extension, etc. We have now mentioned this explicitly in the methods section in lines 536-541.

Line 29 – is there evidence that the rock islands experience more extreme heatwaves? Where is the proof?

See above comment regarding thermal anomalies during ENSO events in the Rock Islands and papers cited in the manuscript.

NA

Line 62-63: Historically? Or is this post-1997/98 bleaching? Be specific as this matters greatly. I'm guessing species richness was historically less in the Rock Islands, but I could be wrong

Coral general richness has been consistent in the Rock Island reefs over the last ~15 years (2002-2016) (Gouezo et al. 2019). For parts of this sampling period the Rock Islands have greater generic richness than all other sampled sites including outer reefs. Sampling for the study cited in that sentence (Barkley et al. 2016) was conducted in 2010 - 2012, so post-1997/98 ENSO. We have now added further detail on these patterns and updated citations to include Gouezo et al. 2019.

64-68

	Citations Barkley, H. C. & Cohen, A. L. Skeletal records of community-level bleaching in Porites corals from Palau. Coral Reefs 35, 1407–1417 (2016). Gouezo, M. et al. Drivers of recovery and reassembly of coral reef communities. Proceedings of the Royal Society B: Biological Sciences 286, (2019).	
Line 66: Are there really greater anomalies or is the absolute temperature just higher? What are the anomalies? What is the climatology for each site?	See above comment regarding thermal anomalies during ENSO events in the Rock Islands and papers cited in the manuscript.	NA
Line 87-89: Again, it's the temperature anomaly that matters with regards to heat stress, not the absolute temperature.	See above comment regarding thermal anomalies during ENSO events in the Rock Islands and papers cited in the manuscript.	NA
Line 114-115: Are these annual averages? It would be helpful to report out seasonal averages (wet/dry) so that this work can be easily compared to the other work you referenced in marginal, variable environments. Are the Rock Islands hotter/cooler than the other sites in Samoa, FL, and the GBR?	This particular line simply referred to the fact that the entire time series of temperatures for Rock Islands sites is warmer than those from outer reefs (i.e. the warm colored time series lines are all consistently above the cooler colored times series which correspond to the outer reef sites). The word “chronically” has been changed to “consistently” in that sentence to avoid confusion. The temperatures shown in figure 1A represent weekly averages (as stated in the caption). We’ve also added detail to the text to avoid confusion. As the reviewer mentions above, it is the local historical temperature conditions that are of greatest interest in considering the thermal tolerance or bleaching responses of corals. We are not sure what would be added to the manuscript by directly comparing the temperatures in the Rock Islands to other sites. In some cases, the Rock Islands are warmer; in others, they are not. For instance, temperatures in the Rock Islands average above 30C, which is comparable to average temperatures in the highly variable America Samoa pool where Palumbi, Barshis, et al. found corals to have higher thermal tolerance.	132-133

	The daily variability in that pool, though (5-6C), is higher than that seen in the Rock Islands (at most 1.5C), so we do not see direct benefit in directly comparing these two environments. Palau Rock Islands temperatures are warmer on average than FL reefs which are higher latitude, but again these are very different environments and have entirely different species compositions, as Caribbean coral populations are distinct from Pacific ones. The main point of our paper is trying to elucidate the genetic patterns that might be driving differences in the responses between corals in close geographic locations. Hence this is what our discussion centers on. We mention other marginal environments as examples of circumstances that can also help breed particularly resilient or resistant corals; and also because these environments are particularly interesting and useful for understanding the current limits of coral thermal physiology.	
Line 176-77: Do you know how much heat stress there was during past ENSO events? If not, you can't just assume thermal stress was comparable between the inner and outer sites.	See above comment regarding thermal anomalies during ENSO events in the Rock Islands and papers cited in the manuscript.	NA
Line 182-83: Reword	We have reworded this sentence to say RD lineage corals "are confined to two of the Rock Island sites with the lowest pH"	214-217
Line 191-92: thermal anomaly?	The lines mentioned list the results for the differences in stress band between DB and LB lineages. The lineage groupings have corals at multiple sites, so reporting a temperature anomaly between the groups is not entirely straightforward. Barkley 2016 (PhD thesis) had calculated the aggregated temperature anomaly for the Rock Islands during 1998 as 1.66 +/- 0.31 C, while aggregate anomalies for the outer reef region most represented in our data set was about 1.69 +/- 0.18C. These are not entirely representative of our sites, however. As this work was not peer-reviewed or published and the reporting of these aggregated numbers may be rather confusing,	NA

	we opted to not include this.	
Line 186-198: How were these data averaged? What years were averaged? Did you include stress bands in your estimate? Growth data that include recent bleaching years may muddle inshore-offshore comparisons; i.e., if using bleaching data, you don't know what growth is like in contrasting environments during benign conditions. The script could flip	See above response to the earlier comment also asking about how growth data were averaged.	NA
Line 196-98: Where are the bleaching data? There is a lot of discussion of past bleaching, but is there actually bleaching observations that support the inferences from density bands? and Line 205-206: you don't have actual bleaching data that I can see, thus your interpretation is inferred. Thus, this statement is too strong without monitoring data to support it	Bleaching patterns in Palau have been studied extensively both by outside researchers and local scientists at the Palau International Coral Reef Center. We do not claim to have bleaching data in our paper. We cite several papers that have documented differences in bleaching between Rock Island and outer reefs sites in our introduction and throughout our paper. In addition, we point the reviewer to the Barkley and Cohen 2016 paper where a portion of our core data came from. This paper directly correlates observed community bleaching prevalence (measured in Bruno et al. 2001) from the same or very nearby sites where coral cores were later collected, and finds extremely high correlation between bleaching prevalence and observed community bleaching ($R^2 = 0.89$, $p < 0.001$). The reviewer does have a good point in questioning whether the presence of a stress band in a particular core proves that THAT individual colony bleached. In this case, we point the reviewer to other studies by our group, in which we have cored colonies pre and post bleaching and found stress bands as expected during known times of bleaching using massive Porites in the Indo-Pacific (DeCarlo and Cohen 2017, and Barkley et al., 2018). Given these findings and (1) the high correlation of stress band prevalence and measured community bleaching levels in Palau (as well as other areas in the Indo-Pacific), and (2) the presence of stress bands in the years that bleaching would be expected, we feel confident that the stress bands seen in our cores indicate historical bleaching.	NA

	Citations Bruno, J. F., Siddon, C. E., Witman, J. D., Colin, P. L. & Toscano, M. A. El Niño related coral bleaching in Palau, western Caroline Islands. Coral Reefs 20, 127–136 (2001). Barkley, H. C. & Cohen, A. L. Skeletal records of community-level bleaching in Porites corals from Palau. Coral Reefs 35, 1407–1417 (2016). Barkley, H. C. et al. Repeat bleaching of a central Pacific coral reef over the past six decades (1960-2016). Communications Biology 1, Article Number: 177 (2018). DeCarlo, T. M. & Cohen, A. L. Dissepiments, density bands and signatures of thermal stress in Porites skeletons. Coral Reefs 36, 749–761 (2017).	
Line 225: Seems to contradict statement in abstract	This sentence was written in a confusing way and does not add substantially to the paragraph, so we have deleted it.	NA
Line 251 onward: What about nutrients? Co-author has done a lot of work on nutrients and their influences on growth. What is going on with nutrients in the Rock islands? What about salinity? Its lower salinity, higher nutrients, right?	From previous work in Palau, there haven't been particularly strong differences in nutrients found between Rock Island sites and outer reefs (Barkley et al. 2015; Table S1). Rock Island sites do have slightly higher levels of nitrogen (nitrate, nitrite, and ammonium) but not in every case. The Drop Off outer reef site, for instance, has comparable levels of nutrients to several Rock Island sites (and higher than some RI sites). These data are also based on point collections of water samples, and so there may be exaggerated or undetected differences between sites that are not seen with such collection methods. Given these limitations, we have not tried to make any claims about the impact of nutrients on coral growth or other responses. We think such claims should really be based on longer term monitoring of those parameters or on experimental results with nutrient levels as the variable of interest.	320-326

Similarly for salinity, Rock Island sites have average salinities ranging from 32.4 to 33.7, and outer reef sites have salinities ranging from 33.1 to 33.9. So the differences are not particularly stark. Since the Rock islands are surrounded by land, there is of course more freshwater runoff. So it could be that there are salinity differences that are more concentrated to regions where groundwater discharges. This particular question was looked at by Shamberger et al. 2017 in the Risong back lagoon, where they found the salinity differences from groundwater discharge. These were strongest in the middle of the back lagoon at depth (>10 m), where they are not really any corals.

Another potential point of interest in the RI could also be higher availability of organic matter and opportunity for heterotrophy. Prior papers have suggested that corals from nearshore reefs have more energy reserves (Anthony 2006) and in general increased heterotrophy can increase growth (Houlbrèque & Ferrier-Pagès 2009).

While we had briefly alluded to some of these other potential mitigating factors in the Rock Islands, we have now added a bit more context in our discussion on lines 320-326, including mention of nutrients and heterotrophy.

Citations

Anthony, K. R. Enhanced energy status of corals on coastal, high-turbidity reefs. *Marine Ecology Progress Series* **319**, 111–116 (2006).

Houlbrèque, F. & Ferrier-Pagès, C. Heterotrophy in tropical scleractinian corals. *Biol Rev Camb Philos Soc* **84**, 1–17 (2009).

Shamberger, K. E. F., Lentz, S. J. & Cohen, A. L. Low and variable ecosystem calcification in a coral reef lagoon under natural acidification. *Limnology and Oceanography*

	(2017) doi:10.1002/lno.10662. Barkley, H. C. et al. Changes in coral reef communities across a natural gradient in seawater pH. Science Advances 1, e1500328 (2015).	
Line 306: LB only dominate in channel; this suggests it does NOT thrive at highest heat	The LB lineage also occurs at Rock Island sites Risong, Taoch, and Ngermid, where water temperatures are high. In fact, it is the dominant lineage at Taoch and Ngermid in addition to the Mecherchar channel (Fig 2). These results show that the LB lineage does thrive at warm, thermally variable Rock Island sites.	NA

Reviewer #3

Reviewer Comment	Response	Line #s
Rivera et. al present a manuscript focused on coral cryptic diversity and thermal tolerance in Palau. The authors combined a range of genetic and historical data to explore the spatial variation and differences in thermal tolerance among cryptic lineages across Palau. They identify a particular lineage that is more common to the marginal Rock Island (LB) sites and that shows reduced stress levels during previous bleaching events. The results also suggest no habitat effect on growth in this lineage. In general, the manuscript is clear, well-written, and explores a timely topic in conservation biology. I would also like to commend the authors on integrating coral core data into analyses of cryptic genetic diversity and thermal tolerance...very cool and novel. There are, however, points that need major clarification before this manuscript would be suitable for publication.	We thank the reviewer for their positive comments and encouragement!	NA

Assigning lineages: It is unclear how the authors deal with admixed individuals. I would suggest removing them from the analyses. What is the threshold to confidently assign each sample a specific genetic lineage? There seems to be a significant amount of admixture, which may be worth discussing more. Please outline this process clearly in the methods	Samples were assigned to their majority group (>50% ancestry for K=4) as their “final” lineage for further analyses. In the three cases where the majority lineage differed between microsatellite STRUCTURE and RADseq assignments, the RADseq assignment was used (Fig. S2). We had previously included this information under the core/growth section of the methods but not the population structure. We have now included it there as well and have also moved the description of structure and lineage selection up before the PCA and other popgen descriptions since these analyses were performed for the lineage groups. Using a >50% lineage assignment has been previously used in the coral literature (e.g. Fifer et al. 2021 and Rippe et al. 2021 for similar comparisons). Interestingly reviewer #1 commented they are surprised by how little admixture we see. We do discuss some of the admixture as a potential indication of introgression or breeding among lineages (lines 165-173), which is likely given that we more recently cross-fertilized two of these lineages: LB and RD. Citations Fifer, J. E., Yasuda, N., Yamakita, T., Bove, C. B. & Davies, S. W. Genetic divergence and range expansion in a western North Pacific coral. Science of The Total Environment 152423 (2021) doi:10.1016/J.SCITOTENV.2021.152423. Rippe, J. P., Dixon, G., Fuller, Z. L., Liao, Y. & Matz, M. Environmental specialization and cryptic genetic divergence in two massive coral species from the Florida Keys Reef Tract. Molecular Ecology 1–17 (2021) doi:10.1111/mec.15931.	495-496; 545-546; 163-173
Gene flow: The authors touch upon levels of divergence among the lineages, but make no attempt explore genetic structure within each lineage. It seems there are enough samples to estimate gene flow between sheltered and	Porites lobata is a gonochoric broadcast spawner. In April 2022, we conducted the first ex situ spawning, fertilization, and larval rearing of P. lobata. Eggs were zooxanthellate and ~200 µm in diameter. Larvae developed to competency in 2 days, although previous	NA

exposed sites. This information would be very informative in interpreting the results. They mention the results from particle tracking models, but perhaps exploring this with the genetic data might provide some insight? What is the reproductive mode of *P. lobata*? Would we expect high gene flow between habitats? If so, how would this change our interpretations of results. RI islands are not self-recruiting, but select for tough lineages from a common gene pool?

studies (Polato et al. 2010) include observations of *P. lobata* larvae remaining in the water column for much longer periods. Based on its reproductive mode, we would expect high dispersal in *P. lobata*, but this may be restricted by the oceanography of our sites. Rock Island lagoons have long residence times, which may restrict larval dispersal.

Reviewer #1 also suggested looking at gene flow within lineages. Here is our response:

We have looked at the levels of gene flow (using F_{st} and N_m as estimated with F_{st}) within each lineage across sites. For some lineages, the information is not very valuable as there are very few individuals or none of that lineage at several sites.

For the LB lineage, we see the expected pattern where the number of expected migrants is higher between RI sites than between RI and OR sites (and also between OR sites). And these results are consistent between RAD and microsatellite derived values. These patterns are corroborated in PCAs with RI samples clustering more closely together than OR samples.

For the DB lineage, the patterns are a little different. With RAD data there are fewer migrants between RI sites than other pairings. However, there are several sites with only 1 individual, and several sites with fewer than 4 individuals, so these estimates are really unideal. For microsatellite data, we have much better sample sizes. We do still see higher F_{st} , lower migrants between RI sites than between OR sites, but there is still lower migration between OR and RI sites than between RI sites.

With the PI and RD lineages neither the RAD or microsatellite data is really useful, as many sites are not represented in either dataset. The PCAs are also fairly uninformative. See below.

While we appreciate the reviewer's suggestion, we think that the uneven representation of lineages across sites makes relying on any of these F_{st} or N_m estimates a bit questionable. So, we prefer to keep the presentation in our manuscript as is and focus on the thermal tolerance and growth differences between lineages instead of delving into specific demographic estimates.

Citation
 Polato, N. R., Concepcion, G. T., Toonen, R. J. & Baums, I. B. Isolation by distance across the Hawaiian Archipelago in the reef-building coral *Porites lobata*. *Molecular Ecology* **19**, 4661–4677 (2010).

Trade-offs: The authors state no clear trade-offs with gains in thermal tolerance in the LB lineage, although I think this claim may need to

The reviewer is correct that Figure 4 suggests there are trade-offs between thermal tolerance and growth. However, these results must be

346-370

be revisited. My interpretation of the data is that there are apparent trade-offs, and that the thermal tolerance of the LB lineage comes at a cost, which is why they are likely outcompeted on the reef slope. Growth rates (particularly skeletal density) are lower in LB than DB corals. The RI sites are effectively selecting for slow growing, but thermally tolerant corals. No trade-offs with gains in thermal tolerance would manifest as no differences in growth between LB and DB lineage.	considered in the context of environmental conditions at each site. As we have written in the discussion, Rock Island sites have lower pH and aragonite saturation, which are known to impact skeletal density and increase bioerosion. We therefore cannot conclude that there is a trade-off between thermal tolerance and growth in Porites cf. lobata. A more parsimonious explanation is that growth and skeletal density in Rock Island corals was influenced by the pH and aragonite saturation at those sites. It is worth noting that data for skeletal density, calcification, and extension rate for LB corals from Rock Island sites shown in Figure 5 do NOT include individuals from the Mecherchar Channel. We recognized that these individuals were exposed to different environmental conditions than their counterparts at Mecherchar and did not include them in the growth rate analysis to avoid biasing the data. We aimed to be conservative in our interpretation, so rather than stating definitively that there is no trade-off between growth and thermal tolerance, we have written that our study does not show clear evidence for a trade-off because environmental conditions present a more likely explanation for the lower growth observed for Rock Island corals. We welcome further studies that examine this question in more detail!	
I believe that once these points are thoroughly addressed, the manuscript will be suitable for publication. Nice paper. Cheers	We thank the reviewer for their comments.	

Reviewers' comments:

Reviewer #1 (Remarks to the Author):

This is a revised manuscript detailing the bleaching resistance and growth patterns of four 'lineages' of *Porites lobata* across Rock Island and Outer Reef habitats of Palau. I reviewed a previous version of this manuscript. In many ways it is much improved, but I still have major reservations about what we should conclude based on the results of this study. Namely, it is very difficult to reconcile the huge amounts of population differentiation between lineages with the fact that they can be cross-fertilized. This problem, when combined with the sequencing and genotyping technique the study uses (single end RADseq) makes it unclear if the genetic patterns the authors describe are reflecting the actual biology of the organism or technical artifacts.

The author's description of their RAD-seq protocol in their response to reviewers was less than reassuring. It is difficult to guess how many reads in their dataset were potentially PCR duplicates, and how many genotypes are based on these artifacts. This has been a known issue since the initial development of the RADseq protocol, depending on the number of PCR cycles that were used to amplify the library (unreported in the Methods section) 50% or more of the reads from a lane can be removed as PCR artifacts. I also don't find the argument that other studies have used single end sequencing to produce RAD data compelling considering PCR duplicates have been a known problem for many years now. While it might be able to find the $K=4$ groups, I am much more skeptical of its ability to identify admixture between those groups, on which much of the Discussion relies (e.g. will these groups remain separate genetic entities in the future? can they interbreed? Is resilience to heat stress heritable?). I encourage the authors to seriously consider this issue, reporting an F_{ST} of 0.7 between potentially interbreeding lineages demands a biological explanation, and every effort should be made to understand if that value is accurate (even without more sequencing, PCR duplicates should produce patterns like an excess of homozygous genotypes that the authors should have been able to test for using the current dataset). It is difficult to think of a study in marine broadcast spawning populations that has found an F_{ST} value so large between sympatric populations, often F_{ST} values between species that have been isolated for a long time are not that large.

The manuscript ends with a discussion of what lineages might be most preferential for restoration efforts given the patterns that they see in this study. However, if these lineages are admixing is it reasonable to assume they will maintain their ability to withstand heat stress across generations? Again, the lack of specificity here about what a 'population' is, and what keeps them separate, makes any discussion of local adaptation between lineages or environments difficult to interpret. Especially because there appears to be genetic differentiation between the Rock Islands and Outer Reef in some lineages but not others (although that is much less obvious in the microsatellite data provided in the response to reviewers), seemingly countering the argument that long retention time in the Rock Islands leads to reduced gene flow and increased local adaptation, which should produce the same patterns in all lineages if they are spawning at the same time (an aside, did the authors try running those Rock Island vs Outer Reef PCAs with all lineages together and looking for a Rock Island vs Outer Reef signal in PCs 3 and onward? If these are all a single species, grouping all colonies might allow the authors to overcome the sample size issues they raise in the response to reviewers). The authors could have also checked for higher rates of inbreeding in the Rock Islands to address this, which wouldn't be so reliant on having polarized groups to see patterns in the PCA when analyzing each lineage separately.

Finally, given that the microsatellite dataset appears to have many more individuals that were successfully genotyped, I would be interested to know if the growth/bleaching patterns described in the manuscript remain the same if individuals were classified into the four groups just by their microsat genotypes. This would presumably lead to more RI vs OR contrasts in the 4 lineages, correct? If that is the case, it might allow the authors to begin disentangling the effect of environments vs. heritable genetic variation in leading to these phenotypes.

Minor Considerations:

Consider putting a site-by-site F_{ST} table for each lineage in the supplemental, I found myself looking for one at several points while reading the study. Also, the authors hint at an isolation by distance population genetic signal for lineages that are found across the study area, consider adding those plots to the supplemental as well.

Response to reviewers

Reviewer comments	Author response	Line numbers
This is a revised manuscript detailing the bleaching resistance and growth patterns of four 'lineages' of Porites lobata across Rock Island and Outer Reef habitats of Palau. I reviewed a previous version of this manuscript. In many ways it is much improved, but I still have major reservations about what we should conclude based on the results of this study. Namely, it is very difficult to reconcile the huge amounts of population differentiation between lineages with the fact that they can be cross-fertilized. This problem, when combined with the sequencing and genotyping technique the study uses (single end RADseq) makes it unclear if the genetic patterns the authors describe are reflecting the actual biology of the organism or technical artifacts.	We thank the reviewer for their comments and respond to their specific points below.	N/A
The author's description of their RAD-seq protocol in their response to reviewers was less than reassuring. It is difficult to guess how many reads in their dataset were potentially PCR duplicates, and how many genotypes are based on these artifacts. This has been a known issue since the initial development of the RADseq protocol, depending on the number of PCR cycles that were used to amplify the library (unreported in the Methods section) 50% or more of the reads from a lane can be removed as PCR artifacts.	Unfortunately, it is impossible to determine the number of PCR duplicates for the RADseq method used in our study (single-enzyme RAD). The mbRAD RADseq method that allows quantification of PCR duplicates is the RADseq, because it has a random-shearing step, and ezRAD can be used with PCR-free kits that eliminate the need for duplicate removals (though it often is not) (Andrews et al. 2014). Some of the options for trying to "remove" or decrease the likelihood of PCR duplicates in RADseq data are not actually likely to improve genotype scoring and can create other kinds of bias	387 527

(Andrews et al. 2014). In response to the reviewer's comment, we have communicated with the company that performed the RADseq analysis in this study (Floragenex) and determined that the number of PCR cycles used was 18. This is now reported in the Methods.

Given that we do not focus on any particular outlier alleles or infer genes that may be under selection across lineages the presence of PCR duplicates in our dataset is unlikely to change our overall conclusions which are based on the all genotype SNPs.

More importantly, a recent study has shown that removal of PCR duplicates from RADseq datasets has little effect on genotypes. Even when 60% of sequenced reads were clones, 95% of calls were unchanged by clone removal (Euclide et al. 2020). Removal of PCR duplicates also did not significantly change estimates of heterozygosity (Euclide et al. 2020). We have now cited this study in our discussion to provide additional context for our results.

References:

Andrews KR, et al. 2014. Trade-offs and utility of alternative RADseq methods: reply to Puritz et al. 2014. *Mol Ecol* 23: 5943-5946

Euclide PT, et al. 2020. Attack of the PCR clones: rates of clonality have little effect on RAD-seq genotype calls. *Mol Ecol Res* 20: 66-78

I also don't find the argument that other studies have used single end sequencing to produce RAD data compelling, considering PCR duplicates have been a known problem for many years now. While it might be able to find the K=4 groups, I am much more skeptical of its ability to identify admixture between those groups, on which much of the Discussion relies (e.g. will these groups remain separate genetic entities in the future? Can they interbreed? Is resilience to heat stress heritable?). I encourage the authors to seriously consider this issue, reporting an FST of 0.7 between potentially interbreeding lineages demands a biological explanation, and every effort should be made to understand if that value is accurate (even without more sequencing, PCR duplicates should produce patterns like an excess of homozygous genotypes that the authors should have been able to test for using the current dataset). It is difficult to think of a study in marine broadcast spawning populations that has found an FST value so large between sympatric populations, often FST values between species that have been isolated for a long time are not that large.

We agree with the reviewer that an F_{ST} value of 0.7 between lineages that appear to interbreed is extremely surprising. We acknowledge that the F_{ST} values may be inflated by PCR duplicates on line 387 of the revised manuscript. In addition, we emphasize that our microsatellite results also recapitulate the same lineages and patterns as our RAD-seq data. So even if our RAD-seq based Fst's are artificially inflated due to PCR duplicates the story remains the same. And again we do NOT incorporate the actual FST value (be it microsatellite or RAD derived) into any downstream analyses.

The key issue here is that the range of F_{ST} values reported for our RADseq data are usually associated with reproductively isolated species. In November 2021, we collected an additional set of *Porites cf. lobata* samples from our same study sites in Palau. Using 2bRAD - a different style of RADseq than this study - we calculated F_{ST} values between lineages. The resulting values ranged 0.2 - 0.4, again in a range that is usually associated with different species, though lower than the Fst's reported here. The fact that a different analysis technique on a different set of samples showed the same general pattern provides support for our results. We again emphasize that the actual value of Fst's are not incorporated into any downstream analyses in this manuscript, e.g. we do not try to infer effective population sizes or migration rate from these, largely because we recognize they may be artificially high. The Fst values in this manuscript simply serve to

387
393

highlight the co-occurrence of lineages with different thermal tolerance across the Palauan seascape. Whether F_{st} values are 0.2 or 0.7 does not change our results or conclusions. We are conducting very thorough follow on experiments to determine the interbreed-ability of these lineages as well as the heritability of the thermal tolerance patterns we've observed.

To be thorough, we calculated the expected heterozygosity for each lineage and each site. The results are listed below. Values are similar between lineages, and the range of values for Rock Island and outer reef sites overlap. If there is an excess of homozygous genotypes, the impact appears to be consistent across the dataset, so our results are unlikely to be biased.

LB: 0.172
DB: 0.148
PI: 0.109
RD: 0.134

Drop Off (OR): 0.158
Helen (OR): 0.234
Kayangel (OR): 0.219
Mecherchar (RI): 0.199
Mecherchar Channel (RI): 0.211
Ngelsible (OR): 0.228
Ngerchelong (OR): 0.156
Ngermid (RI): 0.183
Outer Taoch (OR): 0.169
Risong (RI): 0.242
Taoch (RI): 0.214

While there has been extensive discourse in the literature regarding RADseq biases, the only study to our knowledge that empirically addresses the influence of PCR duplicates in RADseq analysis demonstrates very small impacts of

PCR clone removal on genotype calls and homozygosity, labeling both “not a major concern.” The authors state “Additionally, the sheer number of loci used in most RAD-seq studies means that estimates of diversity and population structure (e.g. pairwise F_{ST}) are fairly robust to a small number of misidentified genotypes” (Euclide et al. 2020).

The reviewer is correct that high genetic differentiation in interbreeding lineages demands an explanation. We have in fact planned an experiment for April 2023 (the upcoming coral spawning season in Palau) that will directly address cross-fertilization and thermal tolerance in *P. cf. lobata* larvae. A full investigation of heritability is outside the scope of the current manuscript, but we have acknowledged in line 393 that studies directly addressing reproductive compatibility between lineages are necessary to resolve the paradox presented by our data.

References:

Euclide PT, et al. 2020. Attack of the PCR clones: rates of clonality have little effect on RAD-seq genotype calls. *Mol Ecol Res* 20: 66-78

The manuscript ends with a discussion of what lineages might be most preferential for restoration efforts given the patterns that they see in this study. However, if these lineages are admixing is it reasonable to assume they will maintain their ability to withstand heat stress across generations? Again, the lack of specificity here about what a 'population' is, and what keeps them separate, makes any discussion of local adaptation between lineages or environments difficult to interpret. Especially because there appears to be genetic differentiation between the Rock Islands and Outer Reef in some lineages but not others (although that is much less obvious in the microsatellite data provided in the response to reviewers), seemingly countering the argument that long retention time in the Rock Islands leads to reduced gene flow and increased local adaptation, which should produce the same patterns in all lineages if they are spawning at the same time.

We agree that what constitutes a "population" is not clearly defined in this study, precisely because this study has uncovered cryptic genetic variation that was not well understood in this species/species complex previously. We see this as a strength and not a drawback of this study, and a finding that opens up exciting new avenues for investigation.

A full investigation of the heritability of thermal tolerance in *Porites* cf. *lobata* is outside the scope of this manuscript. That said, thermal tolerance does appear to be highly heritable in corals, as shown by numerous studies (e.g., Dixon et al. 2015, Kenkel et al. 2015, Dziedzic et al. 2017, Quigley et al. 2020, Bairos-Novak et al. 2021). We have now cited these references in the discussion for additional context.

The fact that lineages can interbreed is fascinating and absolutely deserves further study (which we are doing) as it may mean that there are other barriers to inbreeding (such as reduced offspring fitness in later life stages) that would be fundamental to understanding the dynamics of this important reef system.

In regards to the reviewers last point here, we disagree that one should necessarily expect to see the same patterns of OR/RI differentiation across all lineages. The lineages are not all equally prominent across habitats and as spawning and dispersal in broadcasters is largely stochastic the statistical expectations of dispersal would be different for a less abundant lineage vs a more common one if the distances

traveled are longer. Even if they all spawn concurrently and are able to interbreed, the dispersal potential of a larva spawned on the outer reef is much higher and more variable as currents on the outer reefs are more subject to weather changes in current direction and may disperse more widely (and there are more larvae to begin with hence more that may survive longer distances). Lineages that are more prominent in the Rock Islands would be less likely to make it out of the convoluted Rock Island habitats and therefore, even if they are present on the Outer Reefs, recruitment to/from the Rock Islands to the outer reefs may not occur successfully every year. In addition, there is a higher statistical chance that lineages most prominent on the Outer Reefs may disperse into the Rock Islands since the Outer Reef habitats are **much** larger and have many more coral colonies producing offspring. The fact that we don't see a lot of Outer Reef dominant lineages within the Rock Island may imply that there is some disadvantage they have in surviving in those environments. Which is a neat finding and deserves further study!

References

Dixon GB, et al. 2015. Genomic determinants of coral heat tolerance across latitudes. *Science* 348: 1460-1462

Kenkel CD, et al. 2015. Heritable differences in fitness-related traits among populations of the mustard hill coral, *Porites astreoides*. *Heredity* 115: 509-516

Dziedzic, et al. 2017. Heritable variation in bleaching responses and its functional genomic basis in

	reef-building corals (Orbicella faveolata). Mol Ecol 28: 2238-2253 Quigley, et al. 2020. Genome-wide SNP analysis reveals an increase in adaptive genetic variation through selective breeding of coral. Mol Ecol 29: 2176-2188 Bairos-Novak, et al. 2021. Coral adaptation to climate change: Meta-analysis reveals high heritability across multiple traits. Global Change Biol 27: 5694-5710	
An aside, did the authors try running those Rock Island vs Outer Reef PCAs with all lineages together and looking for a Rock Island vs Outer Reef signal in PCs 3 and onward? If these are all a single species, grouping all colonies might allow the authors to overcome the sample size issues they raise in the response to reviewers. The authors could have also checked for higher rates of inbreeding in the Rock Islands to address this, which wouldn't be so reliant on having polarized groups to see patterns in the PCA when analyzing each lineage separately.	In response to this comment, we plotted PC2 and PC3 for our samples. PC3 explained a very small proportion of the variability in the data (6.7%). The new plot showed no additional patterns in the data that were not already apparent from the PCA included in the manuscript. There was no grouping of individuals between Rock Island and outer reef sites. Figure: Principal components analysis showing PC2 and PC3.  Looking at PC2 and PC3, as colored by Outer Reef (blue) vs Rock Island (red) does not reveal any signals that would be clearer than the lineage groupings. RI/OR groupings overlap and the distinctions are clearly driven by the differential abundance of the lineages across OR and RI	N/A

habitats.

Nor does PC3 and PC4, which as stated start describing very small amounts of variance.

To explore inbreeding, we calculated F_{IS} for each site. Below, each site is listed with its 95% confidence interval for F_{IS} . The values overlap between Rock Island and outer reef sites, indicating no significant difference. In fact, the lowest F_{IS} value is for Ngermid, a Rock Island site.

Drop Off (OR): 0.402 - 0.418
Helen (OR): 0.619 - 0.641
Kayangel (OR): 0.636 - 0.654
Mecherchar (RI): 0.379 - 0.399
Mecherchar Channel (RI): 0.626 - 0.644
Ngelsible (OR): 0.594 - 0.609
Ngerchelong (OR): 0.584 - 0.616
Ngermid (RI): 0.264 - 0.282
Outer Taoch (OR): 0.464 - 0.483
Risong (RI): 0.629 - 0.643

	Taoch (RI): 0.436 - 0.449 Based on the PCA and inbreeding analyses suggested by the reviewer, it does not appear that there is substantial evidence for differentiation between Rock Island and outer reef sites within any of the lineages. These analyses do not contribute any substantially new information to the study, so we have not included them in the revised manuscript.	
Finally, given that the microsatellite dataset appears to have many more individuals that were successfully genotyped, I would be interested to know if the growth/bleaching patterns described in the manuscript remain the same if individuals were classified into the four groups just by their microsat genotypes. This would presumably lead to more RI vs OR contrasts in the 4 lineages, correct? If that is the case, it might allow the authors to begin disentangling the effect of environments vs. heritable genetic variation in leading to these phenotypes.	While the microsatellite dataset has more samples only a subset of these has core data (68) and all of these samples were also processed for RAD-seq. The core data presented in the manuscript already included ALL RAD-seq and microsatellite genotyped samples that had core data. We show the results below using only the microsatellite data, and microsat assigned lineages (given the reviewer's concerns over PCR duplicates), but the patterns remain the same. This in fact removes samples (12) from the dataset and does not facilitate any comparisons. The uneven distribution of lineages among sites prevents any deeper analysis of lineage v. site patterns in the present study. We have included a figure here but have not included a new figure in the manuscript or supplement because the microsatellite analysis did not contribute any substantially new information. Figure: skeletal density, calcification rate, and extension rate of individuals genotyped by	N/A

microsatellites and cored. Asterisks indicate significant differences between individuals from Rock Island v. outer reef sites within a lineage.

Consider putting a site-by-site FST table for each lineage in the supplemental, I found myself looking for one at several points while reading the study.

A site-by-site Fst table for each lineage is only informative for certain pairwise site comparisons, as the distribution of the lineages is highly uneven across sites. Given these limitations we did not include it in the original manuscript.

Per the reviewer's request though, a site-by-site Fst table for each lineage has now been included as Table S3 in the supplement.

Table S3

Also, the authors hint at an isolation by distance population genetic signal for lineages that are found across the study area, consider adding those plots to the supplemental as well.

The reviewer may have misunderstood. The comment about isolation by distance, which appears in line 171 of the manuscript, does not refer to an isolation-by-distance analysis that our team conducted. Rather, the comment is merely pointing out that the general pattern of PI individuals being common on Palau's northern barrier reef and at Helen, which is fairly isolated from the other sites, along with the high F_{ST} values between PI and the other lineages, seems to support a general isolation-by-distance concept. In other words, the lineage with the highest F_{ST} values compared to other lineages is common at the most distant sites in our study area. The isolation of the outermost barrier reef sites could influence the differentiation of the PI lineage.

An isolation by distance analyses for the PI lineage (which the sentences the reviewer refers to was discussing) would not be particularly informative as there is not sufficient distribution or sample sizes in various sites to adequately model within or between site variances.

If using the microsatellite data we have the following sample distribution:

Site	
Drop_Off	3
Helen	7
Kayangel	2
Mecherchar_Channel	0
Melekeok	3
Ngelsible	0
Ngerchelongs	3
Risong	0
Taoch	2

N/A

	For RAD Snps: SiteDrop_Off0Helen1Kayangel1Mecherchar_Channel1Melekeok0Ngelsible3Ngerchelong3Risong3Taoch2 This statement was actually added to the manuscript to address the reviewer's prior comment that such high Fst values required an explanation. This is one possible confounding factor that we are discussing as part of more thoroughly addressing the various potential drivers of the high Fst we see in this system.	Site		Drop_Off	0	Helen	1	Kayangel	1	Mecherchar_Channel	1	Melekeok	0	Ngelsible	3	Ngerchelong	3	Risong	3	Taoch	2	
Site																						
Drop_Off	0																					
Helen	1																					
Kayangel	1																					
Mecherchar_Channel	1																					
Melekeok	0																					
Ngelsible	3																					
Ngerchelong	3																					
Risong	3																					
Taoch	2																					

REVIEWERS' COMMENTS:

Reviewer #1 (Remarks to the Author):

This is the third version of this study that I have reviewed. Overall it has improved since the first version and at this point I have no new critiques to add. The biggest challenge for the reader is still to understand what the 'lineages' are, in previous rounds of review the authors have defended their classification and advocated that they are more akin to populations, but in that case what is maintaining boundaries between them that is detectable in a set of genetic markers that are presumably mostly neutral? I won't restate my previous reviews here but many of the points from earlier rounds of review are still open questions in my mind. I'm not claiming to have an answer to this, and I agree that it is a fascinating question, but the authors make conservation recommendations in the penultimate paragraph of the discussion about prioritizing the LB lineage given these results and as a reader it is difficult to determine how credible those recommendations are and if managers should really act on them. The difficulty arises for a couple of reasons. First, the dataset presented here makes it difficult to determine whether the phenotypes that the authors measured are governed by the local environment or underlying genetic variation, as gaps in the data for different lineages can prevent the authors from drawing conclusions about their performance across habitats (e.g. the RD lineage's absence from the Rock Islands where calcification is more difficult) and thus whether they should be candidates for restoration/conservation in other locations. Second, some force is maintaining those boundaries, so can managers depend on that force remaining consistent in maintaining those lineages in the future, or will they shift as environmental conditions shift? In my opinion the answers to those questions should be in place before advocating for focusing conservation efforts on the LB lineage over the rest.

Review Comments

This is the third version of this study that I have reviewed. Overall it has improved since the first version and at this point I have no new critiques to add. The biggest challenge for the reader is still to understand what the ‘lineages’ are, in previous rounds of review the authors have defended their classification and advocated that they are more akin to populations, but in that case what is maintaining boundaries between them that is detectable in a set of genetic markers that are presumably mostly neutral? I won’t restate my previous reviews here but many of the points from earlier rounds of review are still open questions in my mind. I’m not claiming to have an answer to this, and I agree that it is a fascinating question, but the authors make conservation recommendations in the penultimate paragraph of the discussion about prioritizing the LB lineage given these results and as a reader it is difficult to determine how credible those recommendations are and if managers should really act on them. The difficulty arises for a couple of reasons. First, the dataset presented here makes it difficult to determine whether the phenotypes that the authors measured are governed by the local environment or underlying genetic variation, as gaps in the data for different lineages can prevent the authors from drawing conclusions about their performance across habitats (e.g. the RD lineage’s absence from the Rock Islands where calcification is more difficult) and thus whether they should be candidates for restoration/conservation in other locations. Second, some force is maintaining those boundaries, so can managers depend on that force remaining consistent in maintaining those lineages in the future, or will they shift as environmental conditions shift? In my opinion the answers to those questions should be in place before advocating for focusing conservation efforts on the LB lineage over the rest.

Response

We sincerely thank the reviewer for their continued time in improving our manuscript. We are pleased to hear that the reviewer finds the manuscript much improved. To address the reviewer’s final concerns, we have removed the sentences in the discussion that refer to prioritizing any specific lineage for conservation. We now simply state that the LB lineage deserves further study and that understanding how it is able to maintain thermal tolerance across habitats would be highly valuable in gaining a better understanding of how reefs may fare under changing climate conditions. The new paragraph is excerpted below for ease of reference.

The thermal tolerance of LB corals and their consistent growth across habitats have important implications for understanding the potential of corals to withstand future environmental changes. Many reef systems are characterized by variable or warmer thermal regimes across space and through time, regimes that can select for and harbor thermally tolerant genotypes. Additionally, thermal tolerance appears to be highly heritable in reef-building corals^{47,64–67}. Our results demonstrate that warmer environments can serve as breeding grounds for more tolerant corals (e.g. the LB and RD lineages) and that some of these (e.g. the LB lineage) can a) thrive and maintain their tolerance even when they disperse to cooler environments and b) maintain thermal tolerance without growth obvious trade-offs. Based on its high thermal tolerance and apparent lack of trade-offs, the LB lineage may be particularly well-suited to survive through subsequent increases in temperature and declining pH. Gaining a better understanding of the mechanisms behind its resilience can also provide valuable insights into coral physiology and the future of coral reef ecosystems.